# Subcellular Organelle Targeting as a Novel Approach to Combat Tumor Metastasis

**DOI:** 10.3390/pharmaceutics17020198

**Published:** 2025-02-05

**Authors:** Zefan Liu, Yang Liu, Xin Kang, Lian Li, Yucheng Xiang

**Affiliations:** 1Department of General Surgery, First People‘s Hospital of Shuangliu District (West China Airport Hospital of Sichuan University), Chengdu 610200, China; 19138939183@163.com (Z.L.); 19102822120@163.com (Y.L.);; 2Key Laboratory of Drug-Targeting and Drug Delivery System of the Education Ministry and Sichuan Province, Sichuan Engineering Laboratory for Plant-Sourced Drug and Sichuan Research Center for Drug Precision Industrial Technology, West China School of Pharmacy, Sichuan University, Chengdu 610041, China; liliantripple@163.com; 3School of Pharmacy, Chengdu Medical College, Chengdu 610500, China

**Keywords:** tumor metastasis, targeted therapy, subcellular organelles

## Abstract

Tumor metastasis, the spread of cancer cells from the primary site to distant organs, remains a formidable challenge in oncology. Central to this process is the involvement of subcellular organelles, which undergo significant functional and structural changes during metastasis. Targeting these specific organelles offers a promising avenue for enhanced drug delivery and metastasis therapeutic efficacy. This precision increases the potency and reduces potential off-target effects. Moreover, by understanding the role of each organelle in metastasis, treatments can be designed to disrupt the metastatic process at multiple stages, from cell migration to the establishment of secondary tumors. This review delves deeply into tumor metastasis processes and their connection with subcellular organelles. In order to target these organelles, biomembranes, cell-penetrating peptides, localization signal peptides, aptamers, specific small molecules, and various other strategies have been developed. In this review, we will elucidate targeting delivery strategies for each subcellular organelle and look forward to prospects in this domain.

## 1. Introduction

Tumor metastasis refers to the process by which cancer cells detach from the primary tumor and establish new tumor lesions in the microenvironments of distant organs [1,2,3,4]. This is a primary cause of the high mortality rates associated with cancer [5,6]. The formation of tumor metastasis is a multifaceted physiological process. It involves migrating and invading tumor cells into the bloodstream, their subsequent extravasation as circulating tumor cells (CTCs) into new tissue organs, establishing a pre-metastatic niche, and ultimately forming metastatic nodules. The complete dissection of metastatic nodule is inoperable. In addition, the metastasis formation based on CTCs is cellular level. It is extremely challenging to eradicate tumor metastasis due to the fact that the action sites of most chemotherapeutics are subcellular organelles. Thus, organelle-targeted therapies would further amplify the efficacy and reduce the side effects, which is beneficial for anti-metastasis therapy [7,8]. Although researchers have investigated measures to inhibit tumor metastasis, the development and mechanisms are intricate, thus still requiring more effective strategies [9,10,11,12].

With the continuous advancements in medical and biotechnological fields, the emergence of nanomedicine offers newfound hope in treating metastatic tumors [13,14,15,16]. Doxil (liposome-encapsulated doxorubicin) and Abraxane (albumin-bound paclitaxel nanoparticles) are two such nanomedicines that have been approved and extensively utilized for metastatic ovarian and breast cancer treatments [17,18,19,20,21]. Compared to traditional doxorubicin and paclitaxel, these nanomedicines have notably reduced adverse side effects, such as cardiotoxicity or allergic reactions caused by excipients, enhancing patients’ tolerable dosage [22,23,24,25,26]. However, they haven’t significantly improved the five-year survival rate for patients with metastatic or recurrent metastatic tumors [27,28,29]. By modifying carriers with small molecules (e.g., sialic acid) [30], targeting peptides (e.g., K237 peptide) [31,32], or aptamers (EP23 aptamer) [33,34], there is potential to deliver drugs more effectively to metastasis tumor cells, tumor-associated fibroblasts, and macrophages. These targeted nanomedicine systems, akin to “magic bullets”, guide drugs to the target cells; however, they have not brought the anticipated clinical application shift.

This is because, on the one hand, the final action points of most metastatic tumor treatment drugs are located within subcellular organelles [35,36,37]. For instance, doxorubicin, used in the first-line treatment of triple-negative breast cancer, targets topoisomerase II within the cell nucleus, inhibiting DNA replication and transcription [38,39]. Another first-line drug, paclitaxel, binds and stabilizes microtubule structures to halt the cell cycle [40]. Merely reaching the cell surface or depending on diffusion to enter subcellular action points is insufficient for drug efficacy.

On the other hand, the onset and progression of tumor metastasis are intrinsically linked with the structure and function of each subcellular organelle [36,41]. For example, the metastatic process is accompanied by changes in the size of the cell nucleus, which might serve as a physical barrier to tumor cell migration [42]. Mitochondria, while supplying energy for tumor cell migration, can also elevate the expression of pro-migratory or invasive factors under stress conditions [43]. The endoplasmic reticulum (ER) can influence mitochondrial structure and function through ER-related membranes (MAMs) and regulate metastasis-related protein expression via calcium ion transport [44]. The Golgi apparatus aids in expressing and secreting cell factors associated with tumor metastasis, while Golgi-related microtubules assist tumor cells in directed migration and endurance [45,46].

Each subcellular organelle, with its distinct structure, form, and function, provides different targetable features and strategies [47]. For instance, nuclear targeting often involves modification with nuclear localization sequences (NLS) [48]; the mitochondria, having a lipid-friendly and highly negatively charged surface, typically involves hydrophobic, positively charged targeting groups like triphenylphosphonium (TPP) [49]; and the ER and Golgi apparatus frequently utilize targeting to highly expressed receptors on their surfaces [50].

Based on the above, tumor metastasis is a complex process involving multiple subcellular organelles and cellular physiological mechanisms [51]. Subcellular organelle-targeting strategies offer us a precise avenue to combat tumor metastasis, but their potential still needs to be fully realized. Compared with existing literature, this review focus on reviewing the depth exploration of the subcellular targeting mechanisms. Furthermore, during the dynamic cascade process of tumor metastasis, the possible mechanisms involved with subcellular organelles are summarized. This review delves deeply into tumor metastasis processes and their connection with subcellular organelles. In this review, the related literature in medical research databases, including PubMed, Scopus, Web of Science and ScienceDirect, were analyzed. We further elucidate targeting delivery strategies for each subcellular organelle. We then summarize the article and look forward to future developments in this domain.

## 2. Potential Connection Between Subcellular Organelles and Metastasis

The onset of tumor metastasis is closely linked with the structure and function of subcellular organelles [47,52]. Specifically, the nucleus, mitochondria, endoplasmic reticulum, and Golgi apparatus play a central role within cells. Any alterations in these organelles can significantly influence the metastatic capability of tumors. This review will first focus on how these organelles interact with tumor metastasis (Figure 1) [35,53,54].

### 2.1. Nucleus and Metastasis

The nucleus is a crucial organelle, holding genetic information and governing various cellular functions. It comprises the nuclear envelope, nucleoplasm, nucleolus, and chromosomes [55,56]. Moreover, the nucleus is also instrumental in coordinating and orchestrating cellular responses to various external and internal stimuli, which can affect a cell’s fate, differentiation, proliferation, or apoptosis. The connection between the nucleus and tumor metastasis is mainly highlighted in two aspects.

Firstly, the nucleus serves as the regulatory hub for metastasis-related signaling pathways [57,58,59]. Metastasis is mediated by various growth factors and cytokines, operating through multiple signaling pathways. Since the nucleus stores most of a cell’s genetic information, it contains pathways regulating tumor cell metastasis, such as the Wnt, PI3K/AKT/mTOR, and Ras/Raf/MEK/ERK signaling pathways [60,61,62,63,64]. These signaling pathways are intricate and are modulated by various proteins and co-factors. Their deregulation can lead to an aberrant cellular response, promoting tumorigenesis. The nuclear pore complex (NPC), the exclusive gateway between the cytoplasm and nucleus, is one of the largest protein assemblies in the cell, composed of around 30 different proteins called nucleoporins [65]. To initiate transcriptional responses, all metastatic signaling molecules must enter the nucleus through the NPC, suggesting a pivotal role for the NPC in tumor progression mechanisms [57,66]. This nucleocytoplasmic traffic is mediated by specific transport proteins, e.g., importins and exportins, which recognize and bind to signaling molecules, guiding them through the NPC.

Secondly, the size and morphology of the nucleus present physical barriers to the migration and invasion of tumor cells [67,68]. Changes in nuclear size and shape have been diagnostic markers for cancer since the early 19th century [69,70,71]. Alterations in the nuclear architecture can be indicative of the cell’s health, and nuclear irregularities can be a sign of cellular stress or malignancy. During metastasis, the nuclei of various tumor cell types undergo characteristic changes, mostly unrelated to DNA content alterations. Whether these changes are direct outcomes or drivers of metastasis remains unknown [72,73]. Depending on the cancer type, nuclear enlargement and reduction might be associated with metastasis. For instance, small-cell squamous lung cancer and osteosarcoma metastasis relate to nuclear reduction [74,75,76,77,78]. In contrast, it is associated with nuclear enlargement in breast, prostate, and colon cancers, as well as several other malignancies [70,73,79,80,81,82]. This suggests that nuclear morphology could serve as a predictive marker for the metastatic potential of certain tumors.

Being the largest organelle in cells with a diameter ranging from 3 to 15 microns, the nucleus appears relatively large compared to many tiny pores often encountered during physiological tissue migration. As tumor cells migrate through constricted intercellular gaps or capillaries, the size, shape, and rigidity of the nucleus determine whether they can pass through smoothly [83,84]. Furthermore, the physical coupling between the nucleus and the cytoskeleton might impact cancer cell migration [85]. Lamin A and B, nuclear lamina proteins, are critical in determining nuclear shape and stiffness. For example, lamin A has been correlated with nuclear rigidity [86].

Additionally, alterations in nuclear envelope components might influence the mechanical properties of the nucleus, thereby facilitating metastasis [87]. The dynamics of the nuclear–cytoskeletal interface are vital in understanding how cells navigate through various physical barriers during metastasis. For example, softer and multilobulated nuclei might assist cancer cells in passing through intercellular gaps smaller than the nuclear diameter [82].

### 2.2. Mitochondria and Metastasis

While the nucleus acts as the “control center” of the cell, storing and transmitting genetic information to guide cellular functions, mitochondria serves as the “powerhouse”, producing the indispensable energy. Furthermore, mitochondria also possess their DNA, termed mitochondrial DNA (mtDNA), allowing for them to replicate their DNA independently of the nucleus and manufacture some of the essential proteins [88]. This unique attribute underscores the evolutionary significance of mitochondria and hints at their origin as a once-independent organism. Mitochondrial dysfunction significantly promotes cancer metastasis and drug resistance through multiple mechanisms [89]. The mitochondrial fission induced by DRP1 (Dynamin-Related Protein 1) or KRAS (Kirsten Rat Sarcoma Viral Oncogene Homolog) has been reported to promote metastasis in hepatocellular carcinoma and pancreatic cancer, respectively [90,91]. As for drug resistance, the BNIP3L (BCL2/adenovirus E1B 19 kDa interacting protein 3-like)-mediated mitophagy protects glioblastoma from chemotherapy-induced ROS [92]. The genetic mutations in BCL-2 (B-cell lymphoma-2) and IDH2 (Isocitrate Dehydrogenase 2) also contribute to tumor progression [93,94]. Mitochondria play a vital role in tumor metastasis, primarily in the following three areas [95,96].

Firstly, the adenosine triphosphate (ATP) generated by mitochondrial oxidative phosphorylation provides ample energy for tumor cell migration and invasion into the bloodstream [97]. Moreover, mitochondrial respiration consumes oxygen in tumor tissue, leading to a hypoxic environment, up-regulating hypoxia-inducible factor 1α and vascular endothelial growth factor, promoting angiogenesis and metastasis [98,99]. Hypoxia, often encountered in rapidly growing tumors due to inadequate vascular supply, can induce a plethora of cellular responses, promoting tumor survival or leading to tumor cell death [100,101]. However, even under oxygen-sufficient conditions, tumor cells often prefer glycolysis for energy production over mitochondrial oxidative phosphorylation, contrasting with normal cell metabolism [102]. This is the so-called Warburg effect. The reverse Warburg effect is a recent concept, suggesting that surrounding normal cells, like cancer-associated fibroblasts, undergo glycolysis, producing metabolites like lactate and ketones [103]. Cancer cells subsequently take up these metabolites for oxidative phosphorylation, providing energy for migration and invasion, offering a fresh perspective on how cancer cells interact with their microenvironment and harnessing this interaction to support metastasis. The Warburg Effect also represents an abnormal metabolic reprogramming event in cancer. It offers both challenges and unique therapeutic opportunities for targeted delivery. Due to the production of characteristic lactate, the tumor microenvironment is acidic [104]. Thus, pH-sensitive nanocarriers can selectively release payloads in acidic regions, minimizing off-target effects [105]. Moreover, lactate induces immunosuppression by polarizing tumor-associated macrophages [106]. Using anti-lactate dehydrogenase A agents combined with checkpoint inhibitors might reverse the immunosuppression [107].

Secondly, reactive oxygen species (ROS) are the mitochondrial respiratory electron transport chain’s primary byproducts, with mitochondria being ROS production’s principal site [108,109]. Maintaining intracellular ROS balance is among the core functions of mitochondria. Compared to normal cells, tumor cells exhibit elevated ROS levels, enhancing the induction of redox-sensitive proto-oncogenes and pro-metastatic genes like matrix metalloproteinases (MMPs) [110]. MMPs can degrade the extracellular matrix, aiding tumor cells in breaking down the matrix and entering the bloodstream [111,112]. It is noteworthy to mention that while moderate levels of ROS can act as signaling molecules promoting cell survival and proliferation, excessive ROS can be detrimental. Excessive ROS can further damage mitochondria, leading to mitochondrial dysfunction, inducing cell apoptosis, and inhibiting tumor cell metastasis [113,114,115].

Thirdly, mitochondria are vital structures regulating endogenous cell apoptosis, a physiological process in normal cells to remove damaged and unnecessary cells, preserving cellular health. However, this mechanism often becomes disrupted in tumor cells, primarily through up-regulating mitochondrial anti-apoptotic proteins like Bcl-2 and Mcl-1 [116]. The evasion of apoptosis is a hallmark of cancer, allowing for malignant cells to persist and proliferate, leading to tumor growth. Moreover, the expression of proteins associated with cancer metastasis and invasion, such as MMPs and transforming growth factors, is also promoted. Functional mutations or impaired expression of Bax and Bak, along with p53 gene mutations often observed in the metastatic phase of human tumors, further disturb the mitochondrial apoptosis mechanism [117]. These changes closely relate to tumor cell survival, drug resistance, and invasion, playing pivotal roles in cancer onset, progression, and metastasis. Hence, damaging tumor cell mitochondria and disrupting their functions might be effective strategies to inhibit tumor metastasis.

### 2.3. Endoplasmic Reticulum, Golgi Apparatus, and Metastasis

The endoplasmic reticulum–Golgi apparatus is the primary and most widespread intracellular membranous system [118]. They are not only targets for various chemotherapy drugs, but also have significant implications for tumor metastasis due to their morphology, structure, and function [119]. Firstly, the endoplasmic reticulum–Golgi pathway is the primary site for protein synthesis, modification, and processing. Proteins synthesized by ribosomes enter the endoplasmic reticulum for preliminary folding and modifications like glycosylation. They are then transferred to the Golgi apparatus for further modifications and glycosylation before being secreted extracellularly. These protein signals also include metastasis-associated signaling proteins essential for cell–cell communication, extracellular matrix remodeling, angiogenesis, and more. Examples include TGF-β (Transforming Growth Factor-β), VEGF (Vascular Endothelial Growth Factor), MMPs (Matrix Metalloproteinases), collagens, and proteoglycans [120,121,122,123]. Thus, damaging the structure and function of the endoplasmic reticulum or Golgi apparatus, blocking the endoplasmic reticulum-Golgi pathway, can inhibit the secretion of metastasis-associated signals to some extent.

Compared to the Golgi apparatus, ER has unique connections with tumor cell metastasis. Firstly, ER stress plays dual roles in cancer progression. While mild ER stress supports tumor cell survival, persistent or severe stress can trigger apoptosis. However, the sustained activation of ER stress sensors endows malignant cells with greater tumorigenic, metastatic, and drug-resistant capabilities [124]. For example, PERK (Protein Kinase R-like Endoplasmic Reticulum Kinase) buffers protein-folding stress during this increased secretory load and prevents anoikis during epithelial-to-mesenchymal transition-induced loss of cell–cell contact [125]. Consequently, pretreating cells with a PERK inhibitor dramatically reduced in vivo lung metastasis. Thus, modulating ER stress pathways offers a multifaceted approach to disrupt metastatic adaptation. The ER, serving as a primary intracellular calcium ion reservoir, plays a pivotal role in calcium signal transduction within cells [126]. In recent years, an increasing number of studies have suggested an association between ER calcium balance and signaling with tumor cell metastasis. For instance, cell migration is a tightly controlled process that involves the reorganization of the cytoskeleton [127]. Calcium ions act as key modulators in various cellular processes, including the dynamic changes in the cytoskeleton [128]. During cell migration, cells rely on the release of calcium ions from the ER to regulate these processes. When cells are stimulated, the ER releases stored calcium ions, triggering a series of signaling responses. Excessive calcium release or imbalances in calcium can activate cell apoptosis [129]. This is due to the presence of ER-mitochondrial connections, known as MAMs (Mitochondria-Associated ER Membranes), where calcium transfer between the ER and mitochondria occurs. This calcium flux is vital for the survival of tumor cells, as it drives the tricarboxylic acid cycle and the production of mitochondrial substrates essential for nucleotide synthesis and the proper progression of the cell cycle [130]. However, disruptions or excess in these calcium fluxes can lead to imbalances in intracellular calcium levels. This imbalance can damage mitochondria, causing excessive reactive oxygen production, and activate pathways leading to cell apoptosis. Moreover, STIM1 (Stromal Interaction Molecule 1) and Orai1 are both related to intracellular calcium (Ca^2+^) signaling, specifically tied to the activation of calcium release-activated calcium (CRAC) channels (with STIM1 primarily located on the ER membrane and Orai1 mainly on the plasma membrane) [131]. It has been shown that hyperactivity of SOCE caused by the overexpression of STIM1 and Orai1 is linked to increased metastasis in various types of cancers [132].

Unlike the ER, the Golgi apparatus is not only the final checkpoint for post-translationally modified protein secretion, but is also closely related to cell polarization and directed migration [133]. The Golgi apparatus ensures the concentration of proteins or lipids in specific cell membrane areas through directed secretion, thereby promoting cell polarization, directed migration, and metastasis. This directed secretion is related to the position of the Golgi apparatus within the cell and its collaboration with microtubules. In many cells, the Golgi apparatus is located on one side of the nucleus and faces the direction of cell advancement. This specific location and orientation help ensure that the cell’s secretions are correctly delivered to the front of the cell [134]. Meanwhile, microtubules are the primary transport pathways inside cells. The Golgi apparatus interacts with the microtubule organizing center to ensure that secretory vesicles are transported directionally to specific cell regions along the microtubules. The transport of secretory vesicles along the microtubule network depends on motor proteins, such as kinesin and dynein. These proteins can “walk” on microtubules, driving vesicles to move to the front or back of the cell [135]. The directional control of exocytosis by the Golgi apparatus, combined with the vesicle flow to the leading edge of the cell, promotes the directed migration of tumor cells. Therefore, once the Golgi apparatus undergoes functional disorders due to stress, it can inhibit tumor metastasis to a certain extent. Additionally, the Golgi apparatus plays a critical role in cancer metastasis by modulating protein secretion, post-translational modifications, and vesicular trafficking. Dysregulation of these processes enables cancer cells to remodel the extracellular matrix (ECM), evade immune detection, and establish metastatic niches. For example, Golgi-dependent secretion of IL-8 and VEGF fosters an immunosuppressive TME [136]. Moreover, Golgi-derived vesicles might contain oncogenic miRNAs, which can prime distant sites for metastasis [137].

## 3. Subcellular Drug Delivery Systems for Tumor Metastasis Therapy

Currently, a variety of drugs are used to treat metastatic tumors, with many targeting specific organelles within cells [138,139]. For instance, anthracycline drugs like doxorubicin and epirubicin primarily bind to intracellular DNA, forming anthracycline-DNA complexes, and stabilize DNA double-strand breaks induced by topoisomerase II. Lonidamine targets mitochondrial complex II (succinate dehydrogenase) and complex IV (cytochrome c oxidase), inhibiting the electron transport chain [140]. Thapsigargin acts as an endoplasmic reticulum calcium ATPase inhibitor, leading to an increase in calcium concentration within the endoplasmic reticulum, thereby inducing endoplasmic reticulum stress [141]. Monensin disrupts the balance of sodium and hydrogen ions within the Golgi apparatus, interfering with the normal transport and modification of metastasis-related proteins [142]. By designing appropriate drug delivery systems, drugs can be more efficiently delivered to their intracellular sites of action. This enhances therapeutic effects, reduces side effects, and effectively regulates subcellular structures and functions, inhibiting the metastatic process of tumor cells.

Interestingly, by altering the drug’s action site to a location different from its original target through subcellular-targeting delivery systems, the drug’s mechanism of action might change, leading to unexpected pharmacological effects. For example, Xiang et al. found that by changing the action site of doxorubicin, which naturally targets the nucleus, to the endoplasmic reticulum, its cytotoxicity was significantly reduced, but it acquired a new function beneficial for immune activation [143]. Li and colleagues discovered that delivering doxorubicin to mitochondria effectively inhibits the mitochondrial respiratory chain, causing mitochondrial dysfunction. Exploring new pharmacological effects of existing drugs is one of the areas that urgently needs to be explored in the current subcellular-targeting delivery system [144].

The primary strategies for subcellular targeting include modifications with small molecular groups, targeting/transmembrane peptides and aptamers. Additionally, some nanomaterials inherently target-specific subcellular structures, possibly due to compatibility with certain subcellular features. For instance, nanomaterials with a diameter smaller than 9 nm can pass through the nuclear envelope, granting them the ability to target and enter the nucleus [145]. Materials with a high positive charge tend to target mitochondria with a highly negative surface [146]. Polymers rich in anionic groups (e.g., carboxyl groups) can bind to calcium ions in the endoplasmic reticulum through coordination bonds, achieving endoplasmic reticulum lumen targeting [147]. Chondroitin sulfate (CS) can effectively target the Golgi apparatus by interacting with glycoproteins and glycolipids on the Golgi apparatus [148]. This review lists subcellular-organelle-targeting delivery systems used for metastasis treatment in the past five years (Table 1).

### 3.1. Nucleus-Targeted Drug Delivery Systems for Metastasis Therapy

Nuclear targeting in metastasis treatment encompasses a variety of strategies that have exhibited delivery potential in both in vitro and in vivo anti-metastasis [170]. These strategies leverage the natural ability of certain molecules or functional groups to penetrate the nucleus, thus improving the delivery of therapeutic agents directly to the site of action within cells (Figure 2).

#### 3.1.1. TAT Modification

The TAT peptide, with its sequence ‘YGRKKRRQRRR’, is a renowned cell-penetrating peptide (CPP) and stands as a testament to the advances in nuclear drug delivery. Initially identified in the TAT protein of HIV-1—hence its nomenclature—it boasts a remarkable capacity to penetrate cell membranes and seamlessly access cellular interiors. Due to this intrinsic ability, the TAT peptide is acclaimed as a primary tool in molecular delivery, finding significant applications in gene and protein therapy [171].

In the quest for optimized drug delivery, the TAT peptide can be either covalently or non-covalently bound to nanodrug carriers. To illustrate, it is feasible to chemically link the TAT peptide to functional groups present on nanoparticle surfaces, or have it physically adsorbed onto these surfaces. The TAT peptide’s unparalleled proficiency in nucleus targeting can be primarily attributed to its abundant positive charge. This charge profile endows it with the capability to form affinities with the negatively charged cell membrane, which bolsters its cellular ingress. Beyond the cellular entry, the TAT peptide’s capacity to bind nuclear receptors in the nucleus accentuates its nuclear targeting, making it a formidable agent for precise nuclear delivery [149,172].

Underpinning the potential of such nucleus-targeting entities, Bai and his team have unveiled a groundbreaking HDAC-triggered, self-immolation peptide-CPT nanoassembly system that promises precision in nucleus-targeted drug delivery. This ingenious system harnesses the prowess of nuclei-targeting peptides, ensuring the streamlined entry of peptide-CPT nanoassemblies directly into the nuclei of cancer cells. Once ensconced within, the heightened levels of endonuclease HDACs present in these cells initiate a swift drug release, culminating in HDAC-mediated cytotoxic effects. In a bid to enhance this delivery efficacy, the introduction of dual-targeting ligands, such as RGD/EKA and RGD/TAT, has been leveraged [150]. This strategy guarantees a layered targeting approach, commencing from tumor sites, progressing to individual cancer cells, and culminating at their nuclei. In terms of tangible outcomes, this avant-garde drug delivery paradigm has showcased impressive antitumor efficiency in both laboratory settings and animal models. It offers hope in the relentless battle against both primary and metastatic tumors, forging a pivotal link between nuclear targeting and effective metastasis treatment.

#### 3.1.2. NLS Modification

NLS peptides are specific amino acid sequences that direct proteins to the nucleus. These sequences play a pivotal role in cellular function by ensuring that proteins reach their intended nuclear destination. Discovered in the 1980s by Alan Smith and colleagues during investigations into the large T antigen and SV40 virus, they pinpointed specific sequences in these proteins that were instrumental in directing them to the nucleus [173].

The significance of NLS peptides extends to the realm of nanodrug carriers. Here, they are often covalently or non-covalently modified on the carrier surface. This strategic modification aims to boost drug delivery efficiency directly to the nucleus, particularly when the nucleus houses the drug’s primary target [174]. This preferential targeting is facilitated by the ability of NLS peptides to be recognized by specialized proteins within the cell, such as importins. Upon binding to these proteins, NLS peptides form a complex that transits to the nucleus via the nuclear pore complex. This intricate mechanism acts as a gatekeeper, ensuring that only those molecules or carriers adorned with an NLS can access the nucleus, highlighting the precision and specificity of nuclear targeting. Both NLS and TAT cell-penetrating peptides are popular tools for drug delivery, each exhibiting distinct advantages and challenges in application and efficacy. A primary strength of NLS-modified molecules is their specific targeting towards the cell nucleus, ensuring the precise delivery of drugs or other active compounds directly to their nuclear destination. This is particularly critical for therapeutics designed to act within the cell’s nucleus. However, this specificity can also be seen as a limitation, as molecules modified with NLS might not enter cells as readily as those incorporated with TAT cell-penetrating peptides. On the other hand, the TAT peptide is renowned for its broad cellular penetration capabilities, making it versatile for a range of applications. Yet, it may need more nuclear-targeting precision inherent to NLS, potentially limiting its effectiveness in certain scenarios.

Highlighting the profound connection between nuclear targeting and metastasis suppression, Luo and colleagues unveiled a nucleus-targeted drug delivery system specifically tailored for tackling colorectal cancer (CRC) liver metastasis [151]. Their innovative targeting mechanism employed cationic PPMS for DNA condensation, HA fused with PEG for robust stabilization and delivery, and, crucially, NLS peptides for efficient intracellular transit. This delivery system’s success was evidenced by its potent suppression of metastasis, as seen in its ability to inhibit tumor growth and liver metastases in a mouse model. Furthermore, both in vitro and in vivo tests underscored the efficacy of this approach, where the up-regulation of miR-214 emerged as a central player. This groundbreaking research illuminates the potential of using this vector for the systemic delivery of pcDNA-miR-214, presenting a promising therapeutic frontier against CRC liver metastasis. Liu et al. reported a multi-functionalized subcellular-targeting delivery system for efficiently delivering Cas9/sgRNA plasmids to tumor cell nuclei [152]. This system was constructed using protamine-compacted Cas9/sgRNA plasmids, further decorated with peptide and aptamer-conjugated alginate derivatives. Leveraging the nuclear location signal peptide and AS1411 aptamer, which has an affinity for nucleolin in tumor cell membranes and nuclei, the delivery vector specifically targets tumorous cell nuclei, knocking out the PTK2 gene to down-regulate FAK. This genome editing induces mitochondrial-dependent tumor cell apoptosis and suppresses tumor invasion and metastasis. The approach offers a promising strategy for creating multi-functionalized delivery vectors for genome editing.

#### 3.1.3. Aptamer Modification

Aptamers, specific single-stranded DNA or RNA molecules, were first identified by Ellington and Szostak in 1990 by utilizing the SELEX (Systematic Evolution of Ligands by Exponential Enrichment) technique. When contrasted with targeting peptide modifications, aptamers bring forth many benefits, encompassing high specificity, impressive stability, reversibility, and a straightforward synthesis and modification process. Nonetheless, they have limitations such as possible immunogenicity and a reduced in vivo half-life [175,176].

Central to our discussion, the AS1411 aptamer is a guanosine-rich nucleic acid sequence with a special affinity for the Nucleolin protein, found both on the cell surface and importantly within the nucleus [177,178]. By modifying AS1411, it can be steered to target the nucleus due to its pronounced affinity for particular proteins housed therein. This intrinsic binding proficiency allows for the AS1411 aptamer to serve as an effective conduit for delivering drugs or therapeutic agents directly to the nuclei of tumor cells. This specific approach ensures a more pinpointed and potent antitumor intervention.

Building on this concept, Zhao et al. devised a nucleus-targeted photo-immune stimulator (PIS) to bolster antitumor immunity [153]. This PIS, an intricate blend of SAHA-loaded manganese-porphyrin MOF and the AS1411 aptamer, is specifically designed to target the nucleus, initiating chromatin decompaction. This system facilitates photodynamic DNA damage when exposed to laser irradiation, subsequently releasing DNA into the cytosol. This action then sets the DNA/cGAS-STING pathway into motion, working with PDT-induced cellular death to invoke a robust immune response. Notably, the efficacy of this system is evident, as it obstructs tumor progression and metastasis by amplifying the maturation of dendritic cells and encouraging the infiltration of immune cells into tumors, establishing a clear link between nucleus targeting and the mitigation of metastasis.

#### 3.1.4. Other Strategies

Central to many biological processes, the cell nucleus is a prime target for therapeutic interventions, especially in cancer treatment. Nitrogen functional groups are especially adept at nuclear targeting. This is predominantly because they can adopt a positive charge, allowing for them to form electrostatic interactions with the negatively charged components within the nucleus. Furthermore, these functional groups tend to form stable hydrogen bonds with nuclear molecules and may engage in specific interactions with the cell’s nuclear pore complex. Alongside these properties, their pH responsiveness further accentuates nitrogen functional groups as powerful agents for nuclear targeting. Su and colleagues have illuminated this potential by devising a nucleus-targeted drug delivery system [154]. This system harnesses red-emissive carbon quantum dots (CSCNP-R-CQDs), which can infiltrate the nuclei of both standard cancer cells and the more elusive cancer stem cells. A pivotal element in this targeting mechanism is the presence of nitrogen functional groups on the CSCNP-R-CQDs’ surfaces. When this system is paired with the potent drug doxorubicin, there is a significant reduction in the viability of HeLa cells and a commendable eradication of cancer stem cells. The implications of this are profound, suggesting an avenue not only for addressing primary tumors, but also for thwarting metastasis and recurrence—two major challenges in oncology. In in vivo studies, the therapeutic impact of the combined CSCNP-R-CQDs/doxorubicin approach outperformed free doxorubicin, showing no signs of cancer recurrence. This groundbreaking research paves the way for innovative strategies in crafting efficient nano-drug delivery systems, promising a more effective approach to cancer treatment.

Additionally, the nuclear pore complex plays a pivotal role in the selective exchange of molecules between the nucleus and the cytoplasm. Sporting a diameter of approximately 9 nm, these channels are discerning gatekeepers; not every molecule has the privilege to traverse freely. The size and structure of a molecule intricately determine the passage criteria of the nuclear pore. While molecules smaller than 9 nm effortlessly drift through via a straightforward diffusion mechanism, those exceeding this size necessitate either a distinct nuclear transport signal or a union with a nuclear transport receptor to gain entry. This meticulous selectivity acts as a safeguard, ensuring the molecular composition and functionality within the nucleus remain properly orchestrated and modulated. Drawing a profound connection between cellular nuclear targeting and its implications, Zhan and his team unveiled an organelle-mimetic drug delivery system specifically targeting the nucleus, tailored for precision therapy against liver cancer [155]. This ingenious system, christened HpYss, experiences a remarkable metamorphosis from extracellular nanoparticles (50–100 nm) to intracellular nanofibers (4–9 nm) under the governing effects of extracellular alkaline phosphatase (ALP) and intracellular glutathione (GSH). This transformation in both size and form is instrumental in streamlining drug delivery straight to the nucleus. The precise nuclear-targeting of HpYss magnifies apoptosis, significantly augmenting the inhibitory prowess against HepG2 liver cancer cells. Moreover, this strategy manifests a promising potential in obliterating liver tumor proliferation and thwarting the onset of lung metastasis.

### 3.2. Mitochondrial-Targeted Drug Delivery Systems for Metastasis Therapy

Mitochondria have a bilayer membrane structure, consisting of the outer mitochondrial membrane, inner mitochondrial membrane, intermembrane space, and mitochondrial matrix [179]. The permeability of the mitochondrial membrane is very low, making it difficult for drugs to enter the mitochondria and exert their effects. Therefore, specific modifications are required to increase their distribution in the mitochondria. Different drugs have different action sites in the mitochondria. How to efficiently and specifically target drugs to specific sites in the mitochondria is one of the key factors in improving the efficacy of anti-tumor drugs [180,181]. In current research focused on tumor metastasis, mitochondrial targeting strategies predominantly involve the use of high-positive-charge lipophilic groups like triphenylphosphonium (TPP) for modification, as well as the incorporation of mitochondrial targeting sequences (MTS) peptides to ensure precise delivery and enhanced therapeutic efficacy of tumor metastasis (Figure 3).

#### 3.2.1. TPP Modification

The inner mitochondrial membrane possesses a negative membrane potential, typically ranging between −150 to −180 mV. This is attributed to proton pump activity resulting from electron transfer processes within the mitochondrial electron transport chain. This activity causes a lower proton concentration within the mitochondrial matrix compared to its exterior. Triphenylphosphonium (TPP) is a frequently employed targeting moiety for mitochondria [156,157,158,159]. The molecular structure of TPP consists of a phosphorus atom (P) surrounded by three phenyl groups. The phosphorus atom carries a positive charge, and when a hydrogen atom covalently bonded to it is removed (protonated), a positive charge remains on the phosphorus atom. Due to its positive charge, TPP is attracted to the negative membrane potential of the mitochondria. This ensures that TPP and its derivatives accumulate within the mitochondria at concentrations much higher than other cellular regions, achieving selective mitochondrial targeting. Peng and colleagues pioneered a mitochondria-targeted drug delivery system. This system incorporates a TPP moiety on the micelle’s exterior, enhancing tumor lesion distribution, receptor-mediated cellular uptake, and mitochondrial targeting via electrostatic attraction [161]. This system is instrumental in curbing tumor metastasis, underscoring the significance of mitochondrial-targeted delivery for improved therapeutic outcomes in TNBC.

Lonidamine (LND) is a small-molecule drug that has undergone clinical trials, especially when combined with standard chemotherapeutics for various cancers. Its primary action mechanism disrupts mitochondrial function, triggering apoptotic signaling by releasing cytochrome C into the cytoplasm, which subsequently activates caspase-9. This leads to the up-regulation of downstream caspase-3, promoting cell apoptosis. Despite its potential, LND showed limited efficacy in Phase II and Phase III trials for lung cancer, but was deemed safe. The inherent lack of organ specificity in small-molecule drugs like LND calls for its mitochondrial targeting to boost its therapeutic potential and specificity. This ensures the drug accumulates effectively in the desired organelles to exert its antitumor effects. He and colleagues introduced a mitochondria-targeted lonidamine (LND) delivery system, employing a TPP derivative for precise mitochondrial targeting [160]. This targeting mechanism enhances the delivery of LND specifically to the mitochondria. When combined with anti–PD-L1, the system effectively curbs tumor growth and metastasis. The anti-metastatic effects are attributed to the synergistic action of drugs and the triggered immune response against tumor progression.

#### 3.2.2. MTS Modification

In mitochondrial drug targeting, both MTS and TPP modifications have unique strengths and weaknesses. As endogenous sequences, MTS peptides can specifically transport proteins synthesized in the cytoplasmic ribosomes directly to the mitochondria, ensuring a more natural and precise targeting approach. This highlights its advantages in biocompatibility, precise localization, and targeting efficiency that remains unaffected by the mitochondrial membrane potential. However, a significant challenge for MTS peptides is their inherent difficulty in penetrating the cell membrane. Most MTS peptides struggle to cross the cell membrane effectively and can only adhere to its surface, limiting their intracellular delivery applications. On the other hand, TPP, with its positive charge, is naturally drawn to the negative potential of the mitochondria, ensuring its high accumulation within. Yet, due to TPP’s non-peptidic nature, it might not achieve the same targeting specificity as MTS peptides in certain cellular contexts.

MTS peptides used in mitochondrial-targeted drug delivery are endogenous peptides, typically comprising 15–55 amino acids. They can specifically transport proteins synthesized in the cytoplasmic ribosomes to the mitochondria. These MTS peptides usually bind to the mitochondrial import machinery on the outer mitochondrial membrane through amphipathic α-helices. Once inside the mitochondria, MTS peptides can interact with the internal import machinery or remain stationary, selectively transporting proteins to different mitochondrial regions. However, the inherent difficulty of MTS peptides in crossing the cell membrane severely limits their intracellular applications. To address this, researchers found that covalently binding MTS peptides with cell-penetrating peptides (CPP) enhances the cellular uptake capability of MTS peptides, significantly improving their distribution within the mitochondria.

Li and colleagues introduced a mitochondrial-targeted drug delivery system, utilizing the novel hybrid peptide R8MTS, a fusion of cell-penetrating peptide octaarginine (R8) and mitochondrial targeting sequence ALD5MTS. Mitochondria play a pivotal role in cellular energy production and apoptosis regulation [144]. In this study, by specifically targeting mitochondria, P-D-R8MTS amplifies reactive oxygen species (ROS) production, initiating cell apoptosis, and disrupting mitochondrial function, further inhibiting tumor cell proliferation, migration, and invasion. When the drug DOX-R8MTS enters tumor cells, it is released from the HPMA polymer backbone in the acidic lysosomal environment and effectively targets the mitochondria. This specific mitochondrial targeting strategy enhances ROS production, promoting cell apoptosis. Moreover, by disrupting the mitochondria, this drug delivery system inhibits the growth of breast cancer cells and curbs their migration and invasion, as validated in breast cancer 4T1 and MDA-MB-231 cells. In this therapeutic strategy, mitochondria play a central role, serving both as the target for drug delivery and action and as a key mechanism to inhibit tumor progression and metastasis.

Although R8MTS can specifically deliver DOX to the inner membrane and matrix of mitochondria, it remains unclear whether DOX can be released from DOX-R8MTS within the mitochondria. It has been reported that endogenous mitochondrial matrix targeting sequences can be cleaved by matrix-processing peptidases, releasing proteins to function at specific sites. ALD5 is a polypeptide sequence isolated from the MTS of yeast cells, but there is limited research on whether this polypeptide sequence contains peptidase cleavage sites. A deeper understanding of the mechanism of R8MTS within mitochondria, or finding other MTSs that can specifically release drugs in mitochondria, would enhance the therapeutic efficacy of drugs.

#### 3.2.3. Other Strategies

Given the mitochondria’s inherent high negative charge, modifications with other high-positive-charge groups or peptides can effectively target these organelles. This electrostatic attraction ensures a more precise and efficient delivery of therapeutic agents directly to the mitochondria, leveraging the natural charge dynamics to enhance metastasis treatment outcomes.

2-(Dimethylamino)ethyl methacrylate (DEA) is a positively charged small-molecule hydrophilic compound. Its unique tertiary amine group imparts a strong positive charge, enabling it to bind with the mitochondria’s negative charge effectively. Hence, some studies have demonstrated that DEA can specifically target tumor cell mitochondria using this electrostatic attraction, presenting an intriguing direction for tumor therapy. Recognizing the central role of mitochondria in energy provision, up-regulating pro-metastatic factors, and controlling cell-death signaling, Yi and colleagues posited that targeting CPT to mitochondria by DEA targeting could enhance its therapeutic efficacy against metastatic tumors [162]. To achieve this, they developed a 2-(dimethylamino) ethyl methacrylate (DEA)-modified N-(2-hydroxypropyl) methacrylamide (HPMA) copolymer–CPT conjugate (P-DEA-CPT) to guide the mitochondrial accumulation of CPT. This conjugate was designed to swiftly internalize into 4T1 cells, escape lysosomal degradation, and accumulate effectively in the mitochondria. Once in the mitochondria, P-DEA-CPT severely impaired mitochondrial function, leading to increased reactive oxide species (ROS), energy depletion, amplified apoptosis, and the suppression of tumor metastasis.

Deng and colleagues developed an acid-activated mitochondria-targeted drug nanocarrier to precisely deliver nitric oxide (NO) to combat drug resistance and cancer metastasis [163]. The nanocarrier, constructed using α-cyclodextrin (α-CD) and acid-cleavable dimethylmaleic anhydride-modified PEG conjugated mitochondria-targeting peptide, selectively targets mitochondria under tumor extracellular pH conditions. The mitochondrial targeting peptide, PEG-(KLAKLAK)2CGKRK, is designed with specific amino acid sequences that potentially allow for it to interact with and enter mitochondria. While traditional Mitochondrial Targeting Sequences (MTS) primarily serve as targeting agents, this peptide sequence has the added functionality of potentially inducing mitochondrial dysfunction, which could lead to cell death. Its structure also facilitates conjugation with molecules like PEG, possibly improving blood circulation time. However, while its therapeutic potential is promising, there is a need for caution due to possible unintended effects on healthy cells. NO’s interaction with mitochondria induces mitochondrial dysfunction, affecting ATP levels and inhibiting P-glycoprotein-related activities, pivotal in drug resistance. This disruption further hinders the formation of tumor-derived microvesicles, key players in cancer metastasis. Thus, this innovative approach offers a promising strategy for enhancing the therapeutic efficacy of cancer metastasis treatments.

### 3.3. Endoplasmic Reticulum-Targeted Drug Delivery Systems for Metastasis Therapy

Compared to other subcellular organelles like the nucleus and mitochondria, research related to ER-targeted drug delivery is relatively scarce and challenging. This is primarily due to the limited features of the ER that can be exploited for targeting [182,183]. Current ER-targeting strategies are largely confined to ligand modifications, mainly small molecules and short peptides with ER-targeting properties (Figure 4). Modifying p-toluenesulfonamide and pardaxin peptide to delivery systems are effective strategies. These strategies each have their advantages and have made significant progress in the field of anti-tumor metastasis therapy.

#### 3.3.1. p-Toluenesulfonamide Modification

Some small-molecule drugs used clinically can target the ER based on their inherent structural features. Their mechanisms of action typically include the following: (1) Targeting the sulfonylurea receptors on the ER membrane through the sulfonylurea group present in their chemical structure, e.g., glipizide, a common drug for diabetes. (2) Targeting the chloride ion channels of the ER through the chlorine atoms in their structure. (3) Targeting the phosphatidylcholine cytidylyltransferase (a membrane-bound enzyme presents on the ER membrane and key to phosphatidylcholine synthesis) through the zwitterionic groups in their structure. Small-molecule ligands that target the ER are often used to modify drugs or carriers for drug delivery systems. A commonly used small molecule ligand for anti-tumor metastasis therapy is p-toluenesulfonamide. This molecule can bind to sulfonylurea receptors on the ER membrane, achieving ER targeting.

Shen and his team developed an ER-targeted nanodrug delivery system that effectively delivers anti-cancer drugs to the ER, inducing significant ER stress and autophagy [164]. This system uses p-toluene sulfonamide-modified doxorubicin (ED) as an ER stress inducer. The p-toluene sulfonamide guides ED to the sulfonylurea receptors on the ER, cleaving it by esterases, and releasing DOX. They studied the effects of enhancing and inhibiting autophagy on ER-targeted therapy to inhibit tumor metastasis. They found that enhancing autophagy accelerates the degradation of the core protein snail family transcriptional repressor 1 (SNAI1), thereby inhibiting tumor epithelial–mesenchymal transition and metastasis. In contrast, inhibiting autophagy has the opposite effect. ER-targeted therapy combined with autophagy enhancers showed significantly better anti-metastatic effects than with autophagy inhibitors. Thus, the strategy of enhancing autophagy is more effective in anti-metastatic therapy. Zhang and colleagues reported on an endoplasmic reticulum (ER)-targeted drug delivery system named PLGH-T ER [165]. This system is based on a porous organic framework designed for chemiluminescence resonance energy transfer (CRET)-based photodynamic therapy (PDT) and starvation therapy. Once accumulated in the ER, PLGH-T ER activates the conversion of glucose to H_2_O_2_, leading to starvation therapy and subsequently triggering CRET-based-PDT. This generates reactive oxygen species (ROS) specifically within the ER, causing ER stress and amplifying immunogenic cell death (ICD). The induced ER stress and ROS production kill tumor cells and lead to enhanced immune responses. Furthermore, the PLGH-T ER group showed no metastatic nodules in a lung metastasis model, indicating its effectiveness in treating metastatic tumors. In essence, the PLGH-T ER system offers a promising approach for ER-targeted therapy, effectively addressing metastasis by leveraging ER stress and ROS-induced mechanisms.

#### 3.3.2. Pardaxin Peptide Modification

The ER is integral in intracellular protein synthesis, processing, and transport. By exploring protein transport within cells, we have understood more about ER–tropic signal peptides. These ER signal peptides are often modified at the N-terminus of proteins and perform various functions based on their sequences. For instance, the KDEL signal peptide of lumen-binding proteins can retrieve ER-related proteins from the Golgi apparatus, effectively maintaining the protein homeostasis of the ER. Moreover, most functional complexes on the ER membrane’s surface contain the ER-specific di-arginine (RXR) signal peptide. Modifying drugs or delivery systems with ER signal peptides can effectively enhance their distribution in the ER. Pardaxin peptide (GFFALIPKIISSPLFKTLLSAVGSALSSGGQE) is a natural antimicrobial peptide with ER localization ability. Its ER-targeting mechanism might be related to its efficient phosphatidylcholine vesicle pore-forming capability (phosphatidylcholine is the primary glycerophospholipid of the ER, accounting for 60%). Besides its targeting ability, the Pardaxin peptide also has physiological functions like inducing reactive oxygen species production and ER stress. Yin and colleagues developed an innovative ER-targeted delivery system using pardaxin peptide (PAR)-modified cationic liposomes (PAR-Lipo) for antimetastatic treatments [166]. Pardaxin possesses inherent membrane-penetrating properties. By leveraging the unique capabilities of PAR, the liposomes effectively bypass lysosomal capture, directing the encapsulated DNA toward the ER. This strategic intracellular routing ensures DNA protection and promotes its accumulation in the ER, enhancing the binding of Cas9 and sgRNA. In comparison to p-toluene sulfonyl (a previously used modifier), PAR offers the advantage of a non-lysosomal transport pathway, ensuring enhanced DNA protection and a more direct route to the ER. However, while p-toluene sulfonyl provides specificity through binding to sulfonamide receptors on the ER, PAR’s mechanism is primarily based on its membrane-penetrating properties. When applied in vivo, the PAR-Lipo-mediated knockout of the oncogene CDC6 led to a significant reduction in tumor metastasis, highlighting the potential of this system in targeting and inhibiting cancer metastasis.

#### 3.3.3. Other Strategies

Research has shown that the vitamin family can induce ER stress and mediate cell apoptosis by binding and activating ER-related receptors. Vitamin B6 endows the drug delivery system with some ER-targeting ability, which might be related to the vitamin family receptors distributed in the ER. Wang and colleagues developed a unique ER-targeted drug delivery system using tocopheryl DM1 encapsulated in pH low-insertion peptide (pHLIP) anchored vitamin lipid nanovesicles [167]. This system combines the benefits of lipid–drug conjugates and antibody–drug conjugates to target cancer cells specifically. The targeting mechanism is based on the ability of the vitamin group to induce ER stress-mediated apoptosis in various cancer cells by binding or activating receptors in the ER. Furthermore, the pH-sensitive peptide enhances cellular uptake of the synthesized tocopheryl DM1 and directs it to the endoplasmic reticulum, thereby inducing cancer cell apoptosis. In vitro studies demonstrated that these targeting nanovesicles inhibit MCF-7 cell migration in a dose- and time-dependent manner at an acidic pH of 6.5. This inhibition is attributed to the nanovesicles’ ability to recognize the acidic microenvironment of cancer cells and target the ER specifically. The results further showed a significant reduction in colony formation by MCF-7 cancer cells when treated with the targeting nanovesicles.

### 3.4. Golgi Apparatus-Targeted Drug Delivery Systems for Metastasis Therapy

Compared to the endoplasmic reticulum, research related to drug delivery targeting the Golgi apparatus is limited and more challenging (Figure 5) [184,185]. As the transport center of the cell, the Golgi apparatus has an unstable structure, with rapid and frequent transport of its contents. The Golgi-targeted drug delivery system is often based on observations leading to mechanistic studies [168]. Mechanistic studies have found that these nanoparticles often enter the Golgi apparatus by altering endocytosis pathways. The cellular uptake of nanoparticle drug delivery systems is achieved through endocytosis, and different endocytosis pathways determine their intracellular transport. Clathrin-mediated endocytosis tends to transport nanoparticle carriers to endosomes or lysosomes, where they are subsequently degraded in an acidic multi-enzyme environment. Nanoparticles entering cells through macropinocytosis first enter phagosomes, which then fuse with lysosomes and degrade their contents. For endoplasmic reticulum-targeted nanoparticles entering cells through these two pathways, escaping from lysosomes or phagosomes is critical. Unlike clathrin and macropinocytosis pathways, caveolin is a special lipid raft, and its mediated endocytosis tends to transport contents to the Golgi–endoplasmic reticulum network, providing a passive targeting pathway to the Golgi apparatus. As for Golgi apparatus targeting, chondroitin sulfate and cell membrane camouflage strategies are used.

#### 3.4.1. Chondroitin Sulfate Nanoparticles

Chondroitin sulfate (CS) nanoparticles have recently gained prominence as a promising therapeutic agent. In the context of cancer therapy, CS nanoparticles have demonstrated effective targeting of the Golgi apparatus, an essential cellular organelle involved in various signaling pathways associated with metastasis. Li and his team developed a Golgi-targeting prodrug nanoparticle system by synthesizing retinoic acid (RA)-conjugated chondroitin sulfate (CS), termed CS-RA [148]. This system was observed to accumulate in the Golgi apparatus of cancer cells and release RA in an acidic milieu. The nanoparticles were internalized by tumor cells through endocytosis, specifically utilizing a caveolin-dependent endocytosis mechanism. This offers a route to the Golgi apparatus that avoids endolysosomal compartments. The strong binding affinity of the nanoparticles to the Golgi apparatus was attributed to the GalNAc unit composition of CS, facilitating hydrogen bonding with free cysteine residues in GalNAc-T1, a receptor overexpressed in numerous cancer cells. The disruption of the Golgi apparatus’ structure by CS-RA resulted in the inhibition of several metastasis-associated proteins. When combined with paclitaxel (PTX) to produce PTX-CS-RA, this nanoformulation impeded migration, invasion, and angiogenesis in vitro, and suppressed tumor growth and metastasis in vivo. Collectively, these advancements underscore the potential of chondroitin sulfate nanoparticles as a versatile and potent therapeutic tool for various ailments, particularly in metastasis treatment.

#### 3.4.2. Biomembrane-Coated Nanoparticles

The intracellular distribution of biomembrane-wrapped nanoparticles may involve intricate interactions between multiple receptors and ligands. Wang and colleagues extracted endoplasmic reticulum membranes and mixed them with cationic carriers. They fused the two through probe sonication, creating a nanoparticle composite (EhCv NPs). Further mechanistic studies revealed that its intracellular distribution is influenced by SNARE proteins (primarily mediating vesicle fusion and vesicle-target membrane fusion), caveolin-mediated endocytosis, and endoplasmic reticulum-resident proteins (e.g., KDEL signal peptide) activated retrograde transport pathways from the Golgi to the endoplasmic reticulum.

Chen and his team fused tumor cell membranes with erythrocyte membranes to produce fused membrane-wrapped nanoparticles [168]. They also encapsulated monensin (MON) to block nanoparticle expulsion from subcellular organelles, constructing a system that enhances Golgi targeting and retention. Intracellular distribution mechanism studies indicated that the tumor cell membrane can regulate the nanoparticle entry pathway and mediate fusion with the endoplasmic reticulum–Golgi network. On the one hand, tumor cell membrane-wrapped nanoparticles can enter cells through the caveolin pathway, bypassing lysosomal degradation, and making them more inclined to enter the endoplasmic reticulum–Golgi network. On the other hand, both the tumor cell membrane and the endoplasmic reticulum–Golgi network surface contain abundant membrane fusion-related proteins. These proteins primarily mediate biological membrane fusion, further promoting the recognition and binding of tumor cell membrane-wrapped nanoparticles to the endoplasmic reticulum and Golgi apparatus. By examining cell survival rates on epithelial and mesenchymal cells after administration, the release of cytokines in cell culture supernatants, and the migration and invasion of cells, it was demonstrated that Golgi-targeted retention nanoparticles mediate Golgi swelling stress and dysfunction. This can enhance drug accumulation, reduce the secretion of pro-metastatic factors, and inhibit Golgi-related microtubule movement and migration, thereby suppressing the initiation of tumor metastasis. The study revealed that damaging the Golgi function can enhance drug retention, inhibit cell-directed migration, and further expand the application scope of the Golgi-targeted drug delivery system. The research elucidated the potential mechanisms by which enhanced Golgi targeting and retention drug delivery systems inhibit the metastasis initiation process, providing new insights for designing simple and effective metastatic breast cancer treatment strategies.

#### 3.4.3. Platinum Complex

The Golgi-targeting ability of platinum complexes may be associated with their specific interactions with phosphatidylinositol (PI) on the Golgi membrane. PI on the Golgi membrane provides platinum complexes’ locations and anchoring points, promoting their accumulation in the Golgi apparatus. This specific interaction allows for platinum complexes to be efficiently transported to the Golgi apparatus, where they release their payload. Liang and colleagues introduced a novel Golgi-targeted platinum (II) complex (Pt3), which exhibits cytotoxicity to lung carcinoma approximately 20 times higher than cisplatin [169]. This complex induces significant Golgi stress, resulting in the fragmentation of the Golgi structure and the disruption of its functions. Pt3 activates Golgiphagy, but impedes the fusion of autophagosomes with lysosomes, playing a dual role in autophagy regulation. This leads to a loss of proteostasis and triggers apoptotic cell death. Notably, Pt3 is the first to exploit Golgi stress-mediated dual-regulation of autophagy for effective cancer therapy. Moreover, Pt3 demonstrates remarkable anti-metastasis activity, significantly inhibiting the migration ability of A549 cells.

## 4. Future Prospects

Due to the close relationship between the function and structure of organelles and the occurrence and development of cancer metastasis, research on organelle-targeted drug delivery systems has gradually become the focal point of targeted drug development. Modifying targeting groups on the surface of drugs or carriers makes achieving organelle function regulation and efficient targeted drug delivery possible. This strategy has become the core of organelle-targeted drug design and has shown vast potential in tumor metastasis immunotherapy. However, current research on organelle-targeted drugs is still in its early stages, with many scientific challenges awaiting solutions:

(1) Research on targeting individual organelles remains relatively limited, and there is still a gap between this research and clinical applications. Due to multiple barriers at the tissue, cell, and subcellular levels, few drugs are retained in organelles. Furthermore, actively targeted drug delivery systems, especially those modified with peptides and some cell membrane-derived nanomedicines, require strict quality control and safety risk assessments, limiting their clinical applications. Fortunately, recent years have seen the use of targeting peptide Angiopep-2 for drug delivery in treating patients with brain metastases from breast cancer, entering Phase I clinical trials. Additionally, erythrocyte preparations for the treatment of triple-negative breast cancer have entered Phase III clinical trials. These developments provide guidance for the quality assessment of peptide-modified or cell/cell membrane-derived active targeting preparations. Future research needs to construct a more streamlined and efficient ER-targeting system, delving deeper into its biosafety, targeting optimization, and administration methods.

(2) Most research on organelle targeting remains limited to observational studies, lacking in-depth exploration of the targeting mechanism. A deeper understanding of the targeting mechanism will help us better comprehend the interactions between drug delivery systems and cellular biostructures, facilitating the faster clinical application of actively targeted nanomedicines. Although some classic organelle-targeting strategies have clear targeting mechanisms, most related studies remain observational, lacking mechanistic research.

(3) Tumor metastasis is a dynamic cascade process composed of multiple steps, including the following; (i) tumor cells undergoing epithelial–mesenchymal transition (EMT) to transform from a non-invasive epithelial phenotype to a highly invasive mesenchymal phenotype, detaching from the primary tumor tissue and infiltrating the bloodstream; (ii) surviving in the bloodstream as circulating tumor cells (CTCs) and undergoing hematogenous migration; and (iii) CTCs extravasating, settling, and colonizing in the pre-metastatic niche (PMN) of distant organs. Currently, most organelle-targeted drug delivery strategies target only one specific step in the metastasis process, which is insufficient to halt the entire cascade. Therefore, designing and constructing drug delivery systems that target organelles in tumor cells at different metastatic stages is crucial for tumor metastasis treatment.

(4) Compared to the nucleus and mitochondria, targeting drugs to the endoplasmic reticulum (ER) and Golgi apparatus presents greater challenges, primarily due to the following two reasons: (i) Similar composition to the cell membrane. The ER and Golgi apparatus exchange substances and communicate with the cell membrane through secretory vesicles. Their membrane proteins and phospholipid compositions are closely similar to that of the cell membrane, lacking distinctive features for targeted delivery. (ii) Significant variability across different cell types. The structure and function of the ER and Golgi apparatus vary depending on the cell type. For instance, the specialized sarcoplasmic reticulum in cardiomyocytes, which offers potential targeting features, differs from other cells. Currently, there is a lack of universally applicable targeting strategies.

(5) For ER-targeted drug delivery systems, current research mainly focuses on enhancing ER stress using ER-targeting strategies, with relatively fewer studies on suppressing ER stress. The intensity and duration of ER stress determine various cellular fates, such as survival, metastasis, and apoptosis. Most studies explore how to enhance ER stress through ER targeting to promote tumor cell apoptosis, while research on ER targeting to reduce ER stress levels is relatively limited.

(6) There is a lack of research on prolonging drug accumulation in the ER after achieving ER or Golgi targeting for ER or Golgi-targeted drug delivery systems. Unlike other organelles like the nucleus and mitochondria, the ER serves as an intracellular transit station. Although ER-targeting systems can significantly increase its accumulation in the ER in a short time, its retention is not prolonged, with most drugs rapidly expelled from the cell. This is because the ER primarily facilitates intracellular transport, actively participating in various secretion pathways, leading to drugs entering the ER being rapidly transported out of the cell. A few ER-targeting studies only focus on increasing the initial ER localization of drugs, overlooking the issue of drugs being rapidly expelled and not retained in the ER for extended periods.

(7) Artificial intelligence has been used to assist drug design to optimize efficacy and reduce off-target effects. Therefore, using artificial intelligence might also revolutionize subcellular targeting. Artificial intelligence can be used to integrate multi-omics data to identify critical subcellular targets such as organelle-specific proteins or regulatory miRNAs. Furthermore, computational modeling might be able to predict nanoparticle behavior in organelles, simulate receptor-ligand binding conditions, and develop individualized therapy according to personal genetic information.

## 5. Conclusions

While subcellular targeted drug delivery demonstrates considerable therapeutic potential, several critical challenges must be addressed before clinical translation can be fully realized. Firstly, the specificity and reliability of drug delivery systems require further rigorous verification. Secondly, comprehensive biosafety evaluations of both nanocarriers and their pharmaceutical payloads need to be systematically conducted, particularly regarding long-term biocompatibility and off-target effects. Thirdly, challenges persist in optimizing manufacturing processes and ensuring long-term stability during storage under clinical settings. Given the complex process of tumor metastasis, combination therapy may become the focus of current anti-tumor metastasis treatment. Organelle targeting combined with photodynamic therapy and immunotherapy has progressed in basic research. We hope organelle targeting can be combined with other treatment methods for more in-depth study and application in diseases like cancer metastasis. With the continuous advancement in biomedicine, we anticipate the emergence of simpler, more efficient, and safer organelle-targeting strategies, becoming a key direction in the research and application of drugs for tumor metastasis treatment.

## Figures and Tables

**Figure 1 pharmaceutics-17-00198-f001:**
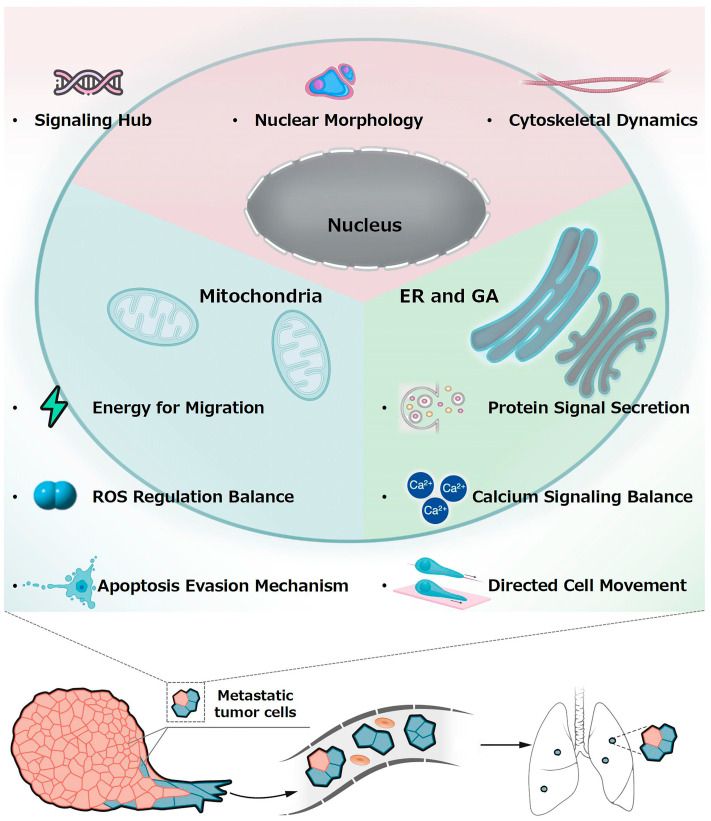
A schematic representation of the interaction between subcellular organelles and tumor metastasis, highlighting the pivotal roles of the nucleus, mitochondria, endoplasmic reticulum (ER), and Golgi apparatus (GA) in influencing metastatic potential.

**Figure 2 pharmaceutics-17-00198-f002:**
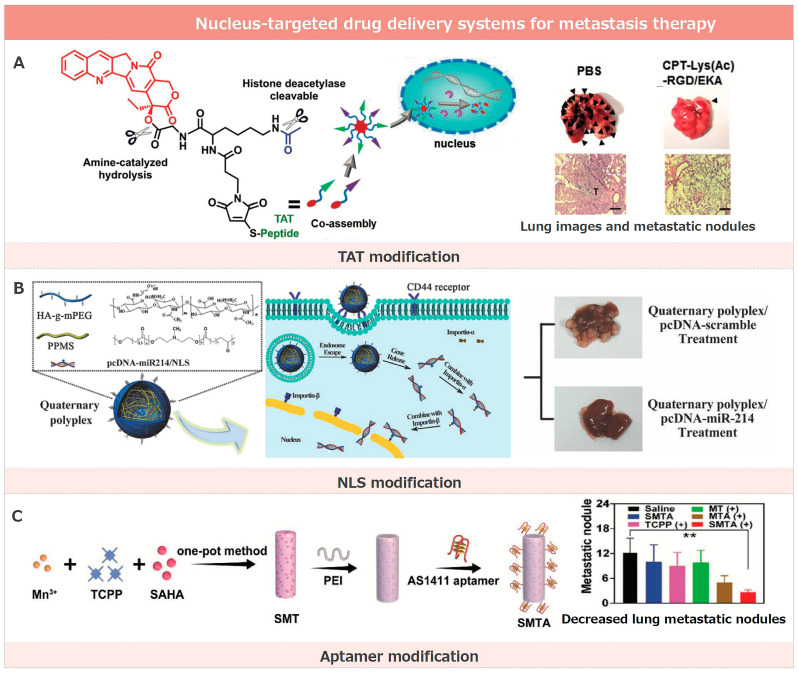
Nucleus-targeted drug delivery systems for metastasis therapy. (**A**) A novel HDAC-triggered nanoassembly system for precise nucleus-targeted drug delivery, utilizing dual-targeting ligands and showcasing significant antitumor efficiency against both primary and metastatic tumors [150]. Reprinted with permission from {H. Bai, H. Wang, Z. Zhou, Y. Piao, X. Liu, J. Tang, X. Shen, Y. Shen, Z. Zhou, Histone Deacetylase-Triggered Self-Immolative Peptide-Cytotoxins for Cancer-Selective Drug Delivery. Adv. Funct. Mater.}. Copyright {2023} Wiley Online Library. (**B**) A nucleus-targeted delivery system effectively combats colorectal cancer liver metastasis, leveraging cationic PPMS, HA with PEG, and NLS peptides, with miR-214 as a central component [151]. Reprinted with permission from {H.-y. Luo, Z. Yang, W. Wei, Y.-q. Li, H. Pu, Y. Chen, H. Sheng, J. Liu, R.-h. Xu, Enzymatically synthesized poly(amino-co-ester) polyplexes for systemic delivery of pcDNA-miRNA-214 to suppress colorectal cancer liver metastasis, J. Mater. Chem. B, 6 (2018) 6365–6376.} Copyright {2018} Royal Society of Chemistry. (**C**) A nucleus-targeted photo-immune stimulator (PIS) induces photodynamic DNA damage, activates the DNA/cGAS-STING pathway, and effectively strengthens immune responses to counteract tumor growth and metastasis [153]. Reprinted with permission from {X. Zhao, K. Zhang, Y. Wang, W. Jiang, H. Cheng, Q. Wang, T. Xiang, Z. Zhang, J. Liu, J. Shi, Intracellular Self-Assembly Driven Nucleus-Targeted Photo-Immune Stimulator with Chromatin Decompaction Function for Robust Innate and Adaptive Antitumor Immunity. Adv. Funct. Mater.}. Copyright {2022} Wiley Online Library. ** *p* ≤ 0.01.

**Figure 3 pharmaceutics-17-00198-f003:**
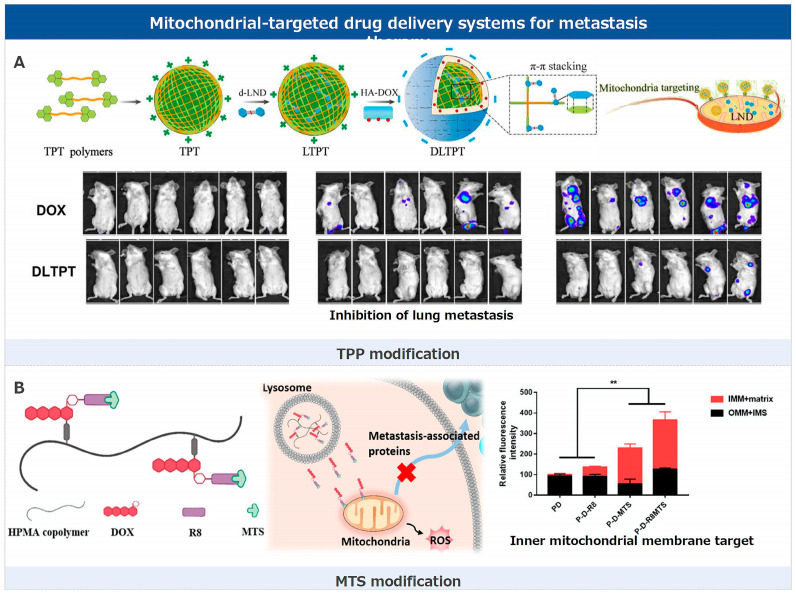
Mitochondria-targeted drug delivery systems for metastasis therapy. (**A**) A mitochondria-targeted lonidamine delivery system was developed using a TPP derivative for precise targeting, which, when combined with anti-PD-L1, effectively inhibits tumor growth and metastasis due to the combined drug action and immune response against tumor progression [160]. Reprinted with permission from {Y. He, L. Lei, J. Cao, X. Yang, S. Cai, F. Tong, D. Huang, H. Mei, K. Luo, H. Gao, B. He, N.A. Peppas, A combinational chemo-immune therapy using an enzyme-sensitive nanoplatform for dual-drug delivery to specific sites by cascade targeting, Science Advances, 7 (2021) eaba0776.}. Copyright {2021} American Association for the Advancement of Science. (**B**) A novel mitochondrial-targeted drug delivery system using the hybrid peptide R8MTS effectively enhances ROS production to promote cell apoptosis, disrupts mitochondrial function, and inhibits breast cancer cell growth and migration, offering a promising therapeutic strategy against tumor progression and metastasis [144]. Reprinted with permission from {Q. Li, J. Yang, C. Chen, X. Lin, M. Zhou, Z. Zhou, Y. Huang, A novel mitochondrial targeted hybrid peptide modified HPMA copolymers for breast cancer metastasis suppression, J. Control. Release, 325 (2020) 38–51.}. Copyright {2020} Elsevier. ** *p* ≤ 0.01.

**Figure 4 pharmaceutics-17-00198-f004:**
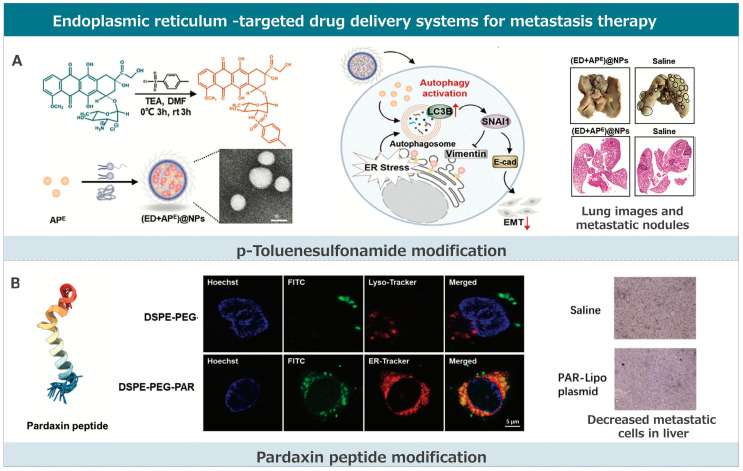
ER-targeted drug delivery systems for metastasis therapy. (**A**) An ER-targeted nanodrug delivery system effectively delivers anti-cancer drugs to induce ER stress and autophagy, with enhanced autophagy found to inhibit tumor metastasis more effectively than inhibited autophagy [164]. Reprinted with permission from {X. Shen, Y. Deng, L. Chen, C. Liu, L. Li, Y. Huang, Modulation of Autophagy Direction to Enhance Antitumor Effect of Endoplasmic-Reticulum-Targeted Therapy: Left or Right?. Adv. Sci. 2023, 10, 2301434.}. Copyright {2023} Wiley Online Library. (**B**) The PAR-modified cationic liposomes offer an efficient ER-targeted delivery system for antimetastatic treatments, bypassing lysosomal capture and ensuring enhanced DNA protection, which, when applied in vivo, significantly reduces tumor metastasis [166]. Reprinted with permission from {H. Yin, X. Yuan, L. Luo, Y. Lu, B. Qin, J. Zhang, Y. Shi, C. Zhu, J. Yang, X. Li, M. Jiang, Z. Luo, X. Shan, D. Chen, J. You, Appropriate Delivery of the CRISPR/Cas9 System through the Nonlysosomal Route: Application for Therapeutic Gene Editing. Adv. Sci. 2020, 7, 1903381.}. Copyright {2020} Wiley Online Library.

**Figure 5 pharmaceutics-17-00198-f005:**
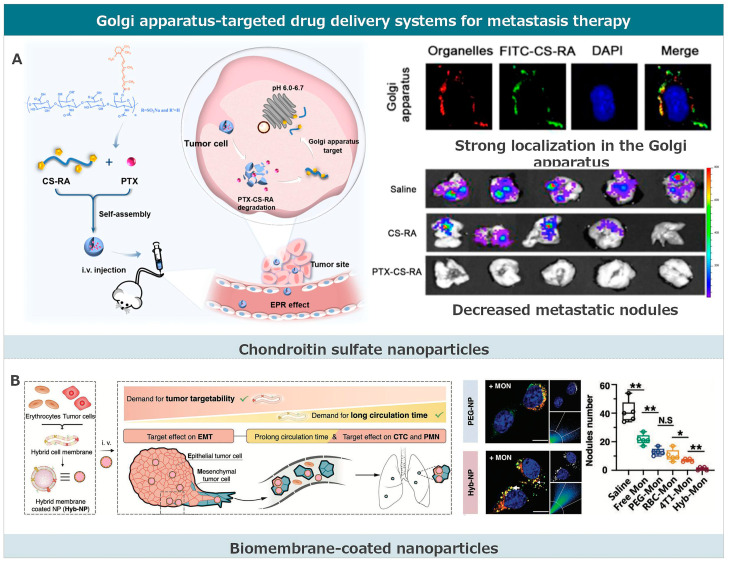
GA–targeted drug delivery systems for metastasis therapy. (**A**) A Golgi-targeted nanoparticle system, CS–RA, accumulates in the Golgi apparatus of cancer cells, releases RA in acidic conditions, and hinders metastasis-associated proteins. When paired with paclitaxel, it effectively curbs tumor growth and spread, showcasing the potential of chondroitin sulfate nanoparticles in metastasis treatment [148]. Reprinted with permission from {H. Li, P. Zhang, J. Luo, D. Hu, Y. Huang, Z.-R. Zhang, Y. Fu, T. Gong, Chondroitin Sulfate-Linked Prodrug Nanoparticles Target the Golgi Apparatus for Cancer Metastasis Treatment, ACS Nano, 13 (2019) 9386–9396.}. Copyright {2019} American Chemical Society. (**B**) Nanoparticles wrapped in tumor cell membranes were developed to target the Golgi apparatus, enhancing drug retention and suppressing tumor metastasis, offering a novel approach for metastatic breast cancer treatment [168]. Reprinted with permission from {L. Chen, P. Jiang, X. Shen, J. Lyu, C. Liu, L. Li, Y. Huang, Cascade Delivery to Golgi Apparatus and On-Site Formation of Subcellular Drug Reservoir for Cancer Metastasis Suppression. Small 2023, 19, 2204747.}. Copyright {2023} Wiley Online Library. * *p* ≤ 0.05, ** *p* ≤ 0.01.

**Table 1 pharmaceutics-17-00198-t001:** The nuclear, mitochondrial, ER, and GA compartment are targeted via different mechanisms. This table summarizes the subcellular-organelle-targeting delivery systems for metastasis treatments in the last five years.

Subcellular Compartment	Targeting Mechanism	Carrier or Material	Drug	Anti-Metastasis Effect	Ref.
**Nuclear**	TAT modification	PEG-PCL NPs	HCPT	Effective antitumor efficacy and inhibition to lung metastasis	[149]
	TAT modification	Self-immolative peptide-camptothecin (CPT) nanoassemblies	CPT	Potent antitumor activity by inhibiting tumor progression and metastasis in breast tumors	[150]
	NLS modification	HA-g-mPEG quaternary polyplexes	pcDNA-miR-214	Highly specific therapeutic approach in the treatment of CRC liver metastasis	[151]
	NLS and AS1411 aptamer modification	Protamine NPs	CRISPR/Cas9 plasmid	Prevents cancer invasion and metastasis in genome-edited cells	[152]
	AS1411 aptamer modification	Metal–organic framework	Vorinostat and photosensitizer TCPP	Significantly enhanced efficacy for inhibiting distant metastasis in several xenograft tumor models	[153]
	Nitrogen groups	Carbon quantum dots	DOX	Therapeutic effect by eliminating CSCs, and shows potential in mediating metastasis	[154]
	Size decreasement (4–9 nm)	Peptide–drug self-assembly NPs	HCPT	Potently abolishing liver tumor growth and inhibiting lung metastasis	[155]
**mitochondrial**	TPP modification	Silica nanoparticles	Catalase	Strong abscopal effect and promising in metastasis inhibition	[156]
	TPP and NLS modification	MSNs	Ce6	Effectively eliminates liver metastasis while sparing hepatocytes	[157]
	TPP modification	Pluronic F127-hyaluronic acid micelles	PTX	Significant antitumor efficacy in a breast cancer-bearing mouse model with lung metastasis	[158]
	TPP modification	MSNs	CO prodrugs	Effective inhibition of tumor growth and metastasis	[159]
	TPP modification	positively charged triphenylphosphonium derivatives particles (LTPT)	lonidamine (LND) dimers (LTPT)	Efficient tumor inhibition and antitumor immune response against tumor metastasis	[160]
	TPP modification	RBC membrane camouflaged cationic micelle	Shikonin	Profound inhibition of lung metastasis in a TNBC mouse model	[161]
	R8-MTS modification	HPMA	DOX	Enhanced reactive oxygen species generation and apoptosis initiation; suppressed migration and invasion of breast cancer 4T1 and MDA-MB-231 cells	[144]
	DEA modification	HPMA	CPT	Anti-metastasis capacity via down-regulation of various pro-metastatic proteins	[162]
	PEG-(KLAKLAK) 2 CGKRK modification	α-cyclodextrin-based NPs	DOX and NO prodrugs	Overcoming drug resistance and cancer metastasis	[163]
**ER**	p-toluene sulfonyl modification	PEG–PLGA NPs	DOX	ER-targeting therapy benefits from the autophagy-enhancing strategy more than the autophagy-inhibiting strategy for antitumor and antimetastasis treatment	[164]
	N-tosylethylenediamine modification	Porous organic framework	GOX and luminol	Effectively activates ICD-induced anti-tumor immunity to hinder the growth of distant and metastatic tumors	[165]
	Pardaxin modification	liposomes	CRISPR/Cas9 system	Down-regulation of cancer cell proliferation and results in fewer metastatic cancer cells in the liver	[166]
	Vitamin E modification	Vitamin lipid nanovesicles	Tocopheryl DM1	Inhibits migration and suppresses tumor growth of metastatic MCF-7 mice	[167]
**GA**	Chondroitin sulfate	Chondroitin sulfate NPs	Retinoic acid	Inhibits migration, invasion, and angiogenesis in vitro and suppresses tumor growth and metastasis in 4T1-Luc-bearing mice	[148]
	Cell membrane camouflaged	Cell membrane camouflaged PLGA NPs	Monensin	Potential therapeutic strategy for cancer metastasis suppression	[168]
	Platinum complex	liposomes	Platinum complex	Migration ability of A549 cells was significantly suppressed in a dose-dependent manner	[169]

## Data Availability

Not applicable.

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
