# Peer review of "Subcellular Organelle Targeting as a Novel Approach to Combat Tumor Metastasis"

_pharmaceutics, 2025, doi:10.3390/pharmaceutics17020198_

Round 1
Reviewer 1 Report
Comments and Suggestions for Authors
The paper addresses a crucial area in cancer research, focusing on organelle-targeted drug delivery as a strategy to combat tumor metastasis and is well written. The discussion covers various organelles, including the nucleus, mitochondria, ER, and Golgi apparatus, and highlights challenges at different stages of tumor metastasis.
While the paper acknowledges challenges in targeting the ER and Golgi apparatus, it could be beneficial to add solutions to these challenges or suggestions. Providing examples of recent advances in targeting these organelles, such as novel ligands, peptides, or nanocarriers with clinical trials will be helpful.
Author Response
Reviewer:
Comments: The paper addresses a crucial area in cancer research, focusing on organelle-targeted drug delivery as a strategy to combat tumor metastasis and is well written. The discussion covers various organelles, including the nucleus, mitochondria, ER, and Golgi apparatus, and highlights challenges at different stages of tumor metastasis.
R:We sincerely thank the reviewer for the insightful comments and positive evaluation of our work. Such constructive feedback from an expert in the field is invaluable for strengthening the translational perspective of this research direction.
Q1:While the paper acknowledges challenges in targeting the ER and Golgi apparatus, it could be beneficial to add solutions to these challenges or suggestions. Providing examples of recent advances in targeting these organelles, such as novel ligands, peptides, or nanocarriers with clinical trials will be helpful.
A.1:Thanks for the comments. The suggested content including solutions to the ER/Golgi apparatus targeting and recent examples have been added correspondingly. Current ER-targeting strategies are largely confined to ligand modifications, mainly small molecules and short peptides with ER-targeting properties. Modifying p-toluenesulfonamide and pardaxin peptide to delivery systems are effective strategies. For specific examples, Shen and his team developed an ER-targeted nanodrug delivery system that effectively delivers anti-cancer drugs to the ER, inducing significant ER stress and autophagy.[1] This system uses p-toluene sulfonamide-modified doxorubicin (ED) as an ER stress inducer. As for Golgi apparatus targeting, chondroitin sulfate and cell membrane camouflage strategies are used. Li and his team developed a Golgi-targeting prodrug nanoparticle system by synthesizing retinoic acid (RA)-conjugated chondroitin sulfate (CS), termed CS-RA.[2] This system was observed to accumulate in the Golgi apparatus of cancer cells and release RA in an acidic milieu. The more detailed information is summarized in Table 1. However, based on our best knowledge, there is no clinical trials of ER or Golgi-targeted therapies for cancer therapy. Pushing the organelle-targeted drug delivery systems to clinical remains challenging.
Reference:
- X. Shen, Y. Deng, L. Chen, C. Liu, L. Li, Y. Huang, Modulation of Autophagy Direction to Enhance Antitumor Effect of Endoplasmic-Reticulum-Targeted Therapy: Left or Right?, 10 (2023) 2301434.
- H. Li, P. Zhang, J. Luo, D. Hu, Y. Huang, Z.-R. Zhang, Y. Fu, T. Gong, Chondroitin Sulfate-Linked Prodrug Nanoparticles Target the Golgi Apparatus for Cancer Metastasis Treatment, ACS Nano, 13 (2019) 9386-9396.
Reviewer 2 Report
Comments and Suggestions for Authors
Subcellular Organelle Targeting as a Novel Approach to Combat Tumor Metastasis
Overview
The manuscript reviews subcellular organelle targeting as a strategy to combat tumor metastasis, emphasizing how organelles like the nucleus, mitochondria, endoplasmic reticulum, and Golgi apparatus influence metastasis and can serve as precise drug delivery targets. It explores various targeting mechanisms, therapeutic strategies, and delivery systems to enhance drug efficacy and reduce metastasis-related complications.
Major Comments
1. Abstract: Provide more specific examples of key findings or novel approaches mentioned in the paper to enhance the clarity and impact of the abstract. It can also highlight the practical implications or breakthroughs of subcellular organelle-targeting strategies.
2. Introduction: Provide a more detailed explanation of the current challenges in tumor metastasis therapy and why subcellular targeting holds promise (A stronger justification of the need for organelle-targeted therapies would provide a more compelling context for the review.). Clearly state the review’s aim and scope, highlighting what the reader can expect from the paper.
3. Emphasize the unique contributions of this review compared to existing literature at the end of introduction.
4. Add simplified diagrams or figures showing the relationship between organelle functions and metastasis processes. Provide a comparative summary of organelles to illustrate their relative importance in metastasis.
5. Nucleus and Metastasis:
6. Specific Case Studies: Include examples of drugs targeting the nucleus that have shown success in preclinical or clinical trials.
7. Nucleus and Metastasis: Include examples of drugs targeting the nucleus that have shown success in preclinical or clinical trials. Streamline the discussion of signaling pathways to focus on their relevance to metastasis.
8. Expanded Discussion on Warburg Effect: Discuss its implications for targeting strategies in greater detail. Include information on recent advancements in mitochondrial drug delivery systems. Include more specific examples of how mitochondrial dysfunction impacts metastasis and drug resistance (This would make the section more actionable and clinically relevant).
9. ER Stress and Cancer: Expand on how ER stress can be modulated therapeutically to combat metastasis. Provide deeper insights into Golgi-related protein secretion and its implications for metastasis. Provide more evidence or examples of successful ER and Golgi-targeted therapies in vivo or in clinical trials (These sections are relatively underexplored, and additional practical insights would strengthen the discussion).
10. Practicality of Strategies: Discuss the challenges in translating these delivery systems from lab to clinic. Highlight novel approaches like nanotechnology and biocompatible materials. Include a comparative analysis of the efficacy and limitations of different targeting strategies (e.g., TPP vs. MTS for mitochondria) (A comparison would help readers evaluate the trade-offs between various delivery methods).
11. Discuss the current limitations of subcellular targeting strategies. Offer clear guidance for future research directions, such as improving the targeting efficiency or addressing immune-related challenges. Highlight emerging technologies like CRISPR, artificial intelligence, or nanotechnology that could revolutionize subcellular targeting (This would add depth and forward-looking relevance to the paper).
Minor Comments
1. Line 16: drug’s.
2. In line reference format. For example, Line 27: with cancer [5,6].
3. Line 68 and 325: NLS expanded as “nuclear localization sequences” and “nuclear localization signal”. Please be consistent.
4. Line 226: Ca2+.
5. Line 286: Table 2? Where is Table 1. Table title or legend?
6. List abbreviations used in the beginning.
7. List future prospects and conclusion separately.
8. Inclusion of more tables would be better.
Remark
Manuscript should be formatted and organized. Check for typo and minor grammatical mistakes.
Should be thoroughly proofread for grammar.
Author Response
Reviewer:
Comments: The manuscript reviews subcellular organelle targeting as a strategy to combat tumor metastasis, emphasizing how organelles like the nucleus, mitochondria, endoplasmic reticulum, and Golgi apparatus influence metastasis and can serve as precise drug delivery targets. It explores various targeting mechanisms, therapeutic strategies, and delivery systems to enhance drug efficacy and reduce metastasis-related complications.
R: Thanks for the comment. We express profound gratitude for the reviewer's astute analysis and generous acknowledgment of our work. The constructive comments help us improve the quality of this review.
Q1: 1. Abstract: Provide more specific examples of key findings or novel approaches mentioned in the paper to enhance the clarity and impact of the abstract. It can also highlight the practical implications or breakthroughs of subcellular organelle-targeting strategies.
A.1: Thanks for the comments. The more specific examples have been added in the revised abstract.
“This review delves deeply into tumor metastasis processes and their connection with subcellular organelles. In order to target these organelles, biomembrane, cell-penetrating peptides, localization signal peptides, aptamers, specific small molecules and various other strategies have been developed. In this review, we will elucidate targeting delivery strategies for each subcellular organelle and look forward to prospects in this domain.”
Q 2: Introduction: Provide a more detailed explanation of the current challenges in tumor metastasis therapy and why subcellular targeting holds promise (A stronger justification of the need for organelle-targeted therapies would provide a more compelling context for the review.). Clearly state the review’s aim and scope, highlighting what the reader can expect from the paper.
A. 2: Thanks for the comments. The corresponding content has been added.[1,2]
“The complete dissection of metastatic nodule is inoperable. Besides, the metastasis formation based on CTCs is cellular level. It is extremely challenging to eradicate tumor metastasis. Due to the fact that the action sites of most chemotherapeutics are subcellular organelles. Thus, organelle-targeted therapies would further amplify the efficacy and reduce the side effects, which is beneficial for anti-metastasis therapy.”
Reference:
- J. Fares, M.Y. Fares, H.H. Khachfe, H.A. Salhab, Y. Fares, Molecular principles of metastasis: a hallmark of cancer revisited, Signal Transduction Targeted Ther., 5 (2020) 28.
- V. Gensbittel, M. Kräter, S. Harlepp, I. Busnelli, J. Guck, J.G. Goetz, Mechanical Adaptability of Tumor Cells in Metastasis, Dev. Cell, 56 (2021) 164-179.
Q. 3: Emphasize the unique contributions of this review compared to existing literature at the end of introduction.
A. 3: Thanks for the suggestion. The corresponding emphasis has been added in the introduction.
“Compared with existing literature, this review focus on reviewing the depth exploration of the subcellular targeting mechanisms. Furthermore, during the dynamic cascade process of tumor metastasis, the possible mechanisms involved with subcellular organelles are summarized.”
Q.4: Add simplified diagrams or figures showing the relationship between organelle functions and metastasis processes. Provide a comparative summary of organelles to illustrate their relative importance in metastasis.
A.4 : Thanks for the suggestion. A simplified figure which shows the relationship between organelle functions and metastasis processes is listed in manuscript. Figure 1 summarizes how the nucleus, mitochondria, ER and GA affect the metastasis. The nucleus might promote metastasis via nuclear morphology and signaling hub. The energy supply, ROS regulation balance and apoptosis evasion mechanism in mitochondria also affect the metastasis. As for the ER and GA, the protein signal secretion, calcium signaling balance and direct cell movement also play a pivotal role in tumor metastasis.
Q.5 Nucleus and Metastasis:
A. 5: In 2.1 Nucleus and metastasis, we summarized that the nucleus is a vital sub-organelle that holds genetic information and controls various cellular functions. The relationship between the nucleus and tumor metastasis is mainly reflected in two aspects. Firstly, the nucleus serves as the regulatory hub for metastasis-related signaling pathways. It contains pathways regulating tumor cell metastasis, such as the Wnt, PI3K/AKT/mTOR, and Ras/Raf/MEK/ERK signaling pathways. Secondly, the size and morphology of the nucleus present physical barriers to the migration and invasion of tumor cells.
Q.6: Specific Case Studies: Include examples of drugs targeting the nucleus that have shown success in preclinical or clinical trials.
A. 6: Thanks for your valuable comments. This review provides a systematic overview of pertinent preclinical research, with a dedicated focus on nucleus-targeting drugs highlighted in Table 1. For example, we mentioned Bai et al. developed a novel HDAC-triggered, self-immolative peptide-CPT nanoassembly system for precise nuclear-targeted drug delivery.[1] This system utilizes nuclei-targeting peptides to facilitate direct entry into cancer cell nuclei, where elevated HDAC levels trigger rapid drug release, inducing cytotoxicity. To enhance delivery, dual-targeting ligands like RGD/EKA and RGD/TAT were incorporated, enabling a multi-level targeting strategy from tumor sites to cancer cell nuclei. This innovative approach demonstrated significant antitumor efficacy in vitro and in vivo, offering potential for treating both primary and metastatic tumors by bridging nuclear targeting with effective metastasis therapy. Luo et al. developed a nucleus-targeted drug delivery system to address colorectal cancer (CRC) liver metastasis, emphasizing the link between nuclear targeting and metastasis suppression.[2] Their system utilized cationic PPMS for DNA condensation, HA-PEG for stabilization, and NLS peptides for intracellular transport. Demonstrating significant antitumor and antimetastatic effects in mouse models, the system's efficacy was validated both in vitro and in vivo, with miR-214 upregulation playing a key role. This study highlights the potential of systemic pcDNA-miR-214 delivery as a promising therapeutic strategy against CRC liver metastasis. Liu et al. developed a multifunctional subcellular targeting system for efficient delivery of Cas9/sgRNA plasmids to tumor cell nuclei.[3] The system combines protamine-condensed Cas9/sgRNA plasmids with peptide- and aptamer-modified alginate derivatives. Utilizing nuclear localization signal peptides and the nucleolin-targeting AS1411 aptamer, the vector specifically delivers plasmids to tumor nuclei, enabling PTK2 gene knockout to downregulate FAK. This genome editing triggers mitochondrial-dependent apoptosis and inhibits tumor invasion and metastasis, demonstrating a promising strategy for multifunctional genome editing delivery. However, based on our best knowledge, there is no clinical trials of nucleus-targeted therapies for cancer therapy. Pushing the organelle-targeted drug delivery systems to clinical remains challenging.
Reference:
[1] H. Bai, H. Wang, Z. Zhou, Y. Piao, X. Liu, J. Tang, X. Shen, Y. Shen, Z. Zhou, Histone Deacetylase-Triggered Self-Immolative Peptide-Cytotoxins for Cancer-Selective Drug Delivery, 33 (2023) 2214025.
[2] H.-y. Luo, Z. Yang, W. Wei, Y.-q. Li, H. Pu, Y. Chen, H. Sheng, J. Liu, R.-h. Xu, Enzymatically synthesized poly (amino-co-ester) polyplexes for systemic delivery of pcDNA-miRNA-214 to suppress colorectal cancer liver metastasis, Journal of Materials Chemistry B, 6 (2018) 6365-6376.
[3] B.-Y. Liu, X.-Y. He, C. Xu, X.-H. Ren, R.-X. Zhuo, S.-X. Cheng, Peptide and Aptamer Decorated Delivery System for Targeting Delivery of Cas9/sgRNA Plasmid To Mediate Antitumor Genome Editing, ACS Appl. Mater. Interfaces, 11 (2019) 23870-23879.
Q.7: Nucleus and Metastasis: Include examples of drugs targeting the nucleus that have shown success in preclinical or clinical trials. Streamline the discussion of signaling pathways to focus on their relevance to metastasis.
A.7: The nucleus and metastasis related examples have been provided in Q.6. Indeed, the nucleus serves as the regulatory hub for metastasis-related signaling pathways. The nucleus contains pathways regulating tumor cell metastasis, such as the Wnt, PI3K/AKT/mTOR, and Ras/Raf/MEK/ERK signaling pathways. However, this review primarily centers on subcellular drug delivery strategies and does not delve into the cell signal transduction pathways associated with tumor metastasis. Modulating the payload in subcellular delivery systems can precisely control the activation/inhibition patterns of the downstream biological pathways.
Q.8: Expanded Discussion on Warburg Effect: Discuss its implications for targeting strategies in greater detail. Include information on recent advancements in mitochondrial drug delivery systems. Include more specific examples of how mitochondrial dysfunction impacts metastasis and drug resistance (This would make the section more actionable and clinically relevant)
A. 8: Thank you for your comment. The suggested content has been added correspondingly.
The Warburg Effect represents an abnormal metabolic reprogramming event in cancer. It offers both challenges and unique therapeutic opportunities for targeted delivery.[1] Due to the production of characteristic lactate, the tumor microenvironment is acidic.[2] Thus, pH-sensitive nanocarriers can selectively release payloads in acidic regions, minimizing off-target effects.[3] Moreover, lactate induce immunosuppression by polarizing tumor-associated macrophages.[4] Using anti- lactate dehydrogenase A agents combined with checkpoint inhibitors might reverse the immunosuppression. [5]
The recent advancements in mitochondrial drug delivery systems have been summarized in Table 1. As for the connection between mitochondrial dysfunction and metastasis/drug resistance [6-11], the specific examples have been added as follows:
“Mitochondrial dysfunction significantly promotes cancer metastasis and drug resistance through multiple mechanisms. The mitochondrial fission induced by DRP1 (Dynamin-Related Protein 1) or KRAS (Kirsten Rat Sarcoma Viral Oncogene Homolog) has been reported to promote metastasis in hepatocellular carcinoma and pancreatic cancer, respectively. As for drug resistance, the BNIP3L (BCL2/adenovirus E1B 19kDa interacting protein 3-like) mediated mitophagy protects glioblastoma from chemotherapy-induced ROS. The genetic mutations in BCL-2 (B-cell lymphoma-2) and IDH2 (Isocitrate Dehydrogenase 2) also contribute to tumor progression.”
Reference:
- Liang, L., Li, W., Li, X., Jin, X., Liao, Q., Li, Y., & Zhou, Y. (2022). ‘Reverse Warburg effect’ of cancer‑associated fibroblasts (Review). International Journal of Oncology, 60, 67.
- Hirschhaeuser, U.G.A. Sattler, W. Mueller-Klieser, Lactate: A Metabolic Key Player in Cancer, Cancer Res., 71 (2011) 6921-6925.
- Wang, D.S. Kohane, External triggering and triggered targeting strategies for drug delivery, Nat. Rev. Mater., 2 (2017) 17020.
- Zhang, X. Zhang, Y. Meng, X. Xu, D. Zuo, The role of glycolysis and lactate in the induction of tumor-associated macrophages immunosuppressive phenotype, Int. Immunopharmacol., 110 (2022) 108994.
- J. Watson, P.D.A. Vignali, S.J. Mullett, A.E. Overacre-Delgoffe, R.M. Peralta, S. Grebinoski, A.V. Menk, N.L. Rittenhouse, K. DePeaux, R.D. Whetstone, D.A.A. Vignali, T.W. Hand, A.C. Poholek, B.M. Morrison, J.D. Rothstein, S.G. Wendell, G.M. Delgoffe, Metabolic support of tumour-infiltrating regulatory T cells by lactic acid, Nature, 591 (2021) 645-651.
- Porporato, P. E., Filigheddu, N., Pedro, J. M. B., Kroemer, G., & Galluzzi, L. (2018). Mitochondrial metabolism and cancer. Cell Research, 28(3), 265-280.
- L. Zhan, H. Cao, G. Wang, Y. Lyu, X. Sun, J. An, Z. Wu, Q. Huang, B. Liu, J. Xing, Drp1-mediated mitochondrial fission promotes cell proliferation through crosstalk of p53 and NF-κB pathways in hepatocellular carcinoma, Oncotarget, 7 (2016) 65001-65011.
- S. Nagdas, J.A. Kashatus, A. Nascimento, S.S. Hussain, R.E. Trainor, S.R. Pollock, S.J. Adair, A.D. Michaels, H. Sesaki, E.B. Stelow, T.W. Bauer, D.F. Kashatus, Drp1 Promotes KRas-Driven Metabolic Changes to Drive Pancreatic Tumor Growth, Cell Reports, 28 (2019) 1845-1859.e1845.
- Y. Sun, G. Zhu, R. Zhao, Y. Li, H. Li, Y. Liu, N. Jin, X. Li, Y. Li, T. Liu, Deapioplatycodin D inhibits glioblastoma cell proliferation by inducing BNIP3L-mediated incomplete mitophagy, Cancer Cell International, 25 (2025) 11.
- R. Singh, A. Letai, K. Sarosiek, Regulation of apoptosis in health and disease: the balancing act of BCL-2 family proteins, Nat. Rev. Mol. Cell Biol., 20 (2019) 175-193.
- J.J. Miller, F. Loebel, T.A. Juratli, S.S. Tummala, E.A. Williams, T.T. Batchelor, I. Arrillaga-Romany, D.P. Cahill, Accelerated progression of IDH mutant glioma after first recurrence, Neuro-oncol., 21 (2019) 669-677.
Q.9: ER Stress and Cancer: Expand on how ER stress can be modulated therapeutically to combat metastasis. Provide deeper insights into Golgi-related protein secretion and its implications for metastasis. Provide more evidence or examples of successful ER and Golgi-targeted therapies in vivo or in clinical trials (These sections are relatively underexplored, and additional practical insights would strengthen the discussion).
A.9: Thanks for the comments. The corresponding content has been added.
Firstly, ER stress plays dual roles in cancer progression. While mild ER stress supports tumor cell survival, persistent or severe stress can trigger apoptosis. However, the sustained activation of ER stress sensors endows malignant cells with greater tumorigenic, metastatic and drug-resistant capabilities.[1] For example, PERK (Protein Kinase R-like Endoplasmic Reticulum Kinase) buffers protein-folding stress during the increased secretory load and prevents anoikis during epithelial-to-mesenchymal transition induced loss of cell-cell contact. Consequently, pretreating cells with a PERK inhibitor dramatically reduced in vivo lung metastasis. Thus, modulating ER stress pathways offers a multifaceted approach to disrupt metastatic adaptation. [2]
Additionally, the Golgi apparatus plays a critical role in cancer metastasis by modulating protein secretion, post-translational modifications, and vesicular trafficking.[3] Dysregulation of these processes enables cancer cells to remodel the extracellular matrix (ECM), evade immune detection, and establish metastatic niches.[4] For example, Golgi-dependent secretion of IL-8 and VEGF fosters an immunosuppressive TME.[5] Moreover, Golgi-derived vesicles might contain oncogenic miRNAs which can prime distant sites for metastasis.[6]
In this review, we have summarized the latest successful ER and Golgi-targeted therapies. Current ER-targeting strategies are largely confined to ligand modifications, mainly small molecules and short peptides with ER-targeting properties. Modifying p-Toluenesulfonamide and pardaxin peptide to delivery systems are effective strategies. For specific examples, Shen and his team developed an ER-targeted nanodrug delivery system that effectively delivers anti-cancer drugs to the ER, inducing significant ER stress and autophagy. This system uses p-toluene sulfonamide-modified doxorubicin (ED) as an ER stress inducer.[7] As for Golgi apparatus targeting, chondroitin sulfate and cell membrane camouflage strategies are used. Li and his team developed a Golgi-targeting prodrug nanoparticle system by synthesizing retinoic acid (RA)-conjugated chondroitin sulfate (CS), termed CS-RA. This system was observed to accumulate in the Golgi apparatus of cancer cells and release RA in an acidic milieu.[8] The more detailed information is summarized in Table 1. However, based on our best knowledge, there is no clinical trials of ER or Golgi-targeted therapies for cancer therapy. Pushing the organelle-targeted drug delivery systems to clinical remains challenging.
Reference:
- X. Chen, J.R. Cubillos-Ruiz, Endoplasmic reticulum stress signals in the tumour and its microenvironment, Nat. Rev. Cancer, 21 (2021) 71-88.
- Y.-x. Feng, E.S. Sokol, C.A. Del Vecchio, S. Sanduja, J.H.L. Claessen, T.A. Proia, D.X. Jin, F. Reinhardt, H.L. Ploegh, Q. Wang, P.B. Gupta, Epithelial-to-Mesenchymal Transition Activates PERK–eIF2α and Sensitizes Cells to Endoplasmic Reticulum Stress, Cancer Discovery, 4 (2014) 702-715.
- B. Chen, G. Kumar, C.C. Co, C.-C. Ho, Geometric Control of Cell Migration, Scientific Reports, 3 (2013) 2827.
- V. Millarte, H. Farhan, The Golgi in cell migration: regulation by signal transduction and its implications for cancer cell metastasis, Scientific World Journal, 2012 (2012) 498278.
- R. Bajaj, A.N. Warner, J.F. Fradette, D.L. Gibbons, Dance of The Golgi: Understanding Golgi Dynamics in Cancer Metastasis, Cells 11 (2022) 1484.
- C.A. Whitehead, A.H. Kaye, K.J. Drummond, S.S. Widodo, T. Mantamadiotis, L.J. Vella, S.S. Stylli, Extracellular vesicles and their role in glioblastoma, Crit. Rev. Clin. Lab. Sci., 57 (2020) 227-252.
- X. Shen, Y. Deng, L. Chen, C. Liu, L. Li, Y. Huang, Modulation of Autophagy Direction to Enhance Antitumor Effect of Endoplasmic-Reticulum-Targeted Therapy: Left or Right?, 10 (2023) 2301434.
- H. Li, P. Zhang, J. Luo, D. Hu, Y. Huang, Z.-R. Zhang, Y. Fu, T. Gong, Chondroitin Sulfate-Linked Prodrug Nanoparticles Target the Golgi Apparatus for Cancer Metastasis Treatment, ACS Nano, 13 (2019) 9386-9396.
Q.10: Practicality of Strategies: Discuss the challenges in translating these delivery systems from lab to clinic. Highlight novel approaches like nanotechnology and biocompatible materials. Include a comparative analysis of the efficacy and limitations of different targeting strategies (e.g., TPP vs. MTS for mitochondria) (A comparison would help readers evaluate the trade-offs between various delivery methods).
A. 10: Thanks for suggestions. The corresponding discussion has been added in conclusion part as follows:
“While subcellular targeted drug delivery demonstrates considerable therapeutic potential, several critical challenges must be addressed before clinical translation can be fully realized. Firstly, the specificity and reliability of drug delivery systems require further rigorous verification. Secondly, comprehensive biosafety evaluations of both nanocarriers and their pharmaceutical payloads need to be systematically conducted, particularly regarding long-term biocompatibility and off-target effects. Thirdly, challenges persist in optimizing manufacturing processes and ensuring long-term stability during storage under clinical settings.”
However, the comparison criteria for different targeting strategies are difficult to evaluate. The efficacy and limitations vary with not only the targeting strategies but also the payload. Thus, case-by-case evaluation might be more reasonable.
Q.11: Discuss the current limitations of subcellular targeting strategies. Offer clear guidance for future research directions, such as improving the targeting efficiency or addressing immune-related challenges. Highlight emerging technologies like CRISPR, artificial intelligence, or nanotechnology that could revolutionize subcellular targeting (This would add depth and forward-looking relevance to the paper).
A.11: Thank you for your suggestions. The current limitations and future prospect of subcellular targeting strategies have been discussed in “4 Future prospect” part. Briefly, it can be concluded into 6 aspects: 1) Research on targeting individual organelles remains relatively limited, and there's still a gap between this research and clinical applications. 2) Most research on organelle targeting remains limited to observational studies, lacking in-depth exploration of the targeting mechanism. 3) Currently, most organelle-targeted drug delivery strategies target only one specific step in the metastasis process, insufficient to halt the entire cascade. 4) Compared to the nucleus and mitochondria, targeting drugs to the endoplasmic reticulum (ER) and Golgi apparatus presents greater challenges due to similar composition to the cell membrane and significant variability across different cell types. 5) For ER-targeted drug delivery systems, current research mainly focuses on enhancing ER stress using ER-targeting strategies, with relatively fewer studies on suppressing ER stress. 6) There’s a lack of research on prolonging drug accumulation in the ER after achieving ER or Golgi targeting for ER or Golgi-targeted drug delivery systems.
The emerging artificial intelligence technology related discussion has been added as follows:
“The artificial intelligence has been used to assist drug design to optimize efficacy and reduce off-target effects. Therefore, using artificial intelligence might also revolutionize subcellular targeting. The artificial intelligence can be used to integrate multi-omics data to identify critical subcellular targets such as organelle-specific proteins or regulatory miRNAs. Furthermore, the computational modeling might be able to predict nanoparticle behavior in organelles, simulate receptor-ligand binding condition and develop individualized therapy according to personal genetic information.”
Q.12: Minor Comments
- Line 16: drug’s.
- In line reference format. For example, Line 27: with cancer [5,6].
- Line 68 and 325: NLS expanded as “nuclear localization sequences” and “nuclear localization signal”. Please be consistent.
- Line 226: Ca2+.
- Line 286: Table 2? Where is Table 1. Table title or legend?
- List abbreviations used in the beginning.
- List future prospects and conclusion separately.
- Inclusion of more tables would be better.
Remark
Manuscript should be formatted and organized. Check for typo and minor grammatical mistakes.
A. 12: Thanks for the constructive suggestions. We have revised the manuscript accordingly and checked the entire manuscript.

Reviewer 3 Report
Comments and Suggestions for Authors
The review article "Subcellular Organelle Targeting as a Novel Approach to Combat Tumor Metastasis" is submitted for review. The title of the review article reflects its content. This article addresses the issue of the relationship between the process of tumor metastasis and subcellular organelles. In addition, the authors explain the targeting strategies for each subcellular organelle.
In the Abstract section, the authors pointed out the relevance of metastatic disease therapy. According to the authors, targeting subcellular organelles offers promising opportunities for improving drug delivery and the effectiveness of metastasis therapy. The Abstract section fully corresponds to the content of the article.
The "Keywords" presented in the article correspond to the content of the article and are necessary.
The introduction briefly presents the mechanisms of metastasis. The authors pointed out the inefficiency of the approaches known in the clinic to treat metastatic disease. The material on the connection between the progression of tumor metastases and the structure and function of each subcellular organelle is very important. The authors formulated the aim of the study clearly. This review delves deeply into tumor metastasis processes and their connection with subcellular organelles. In addition, strategies for targeted drug delivery for each subcellular organelle are covered.
The authors conducted a literature search on the following topics: tumor metastasis and its development mechanisms, modern therapy of metastatic disease, the relationship of the metastatic process with subcellular organelles. It seems to me that the article would be appropriate to reference the medical research databases in which the authors searched for publications necessary for their review article. Are these PubMed, Scopus and/or Web of Science, and possibly other databases? Were there any limitations when searching for publications?
The main search results were analyzed and presented in the following sections of the article: "2. Potential connection between subcellular organelles and metastasis", "3. Subcellular drug delivery systems for tumor metastasis therapy", "4. Future prospect and conclusion". All sections are necessary. The authors clearly, logically and consistently present the structure of organelles (nucleus, mitochondria, endoplasmic reticulum, Golgi apparatus) in cells and their role in metastasis. An analysis of possible approaches to organelle regulation for the correction of metastatic disease was carried out. The discussion of all results is correct using a significant number of publications from various independent research groups. Each section presents an intermediate conclusion. All terms are used correctly.
The authors rightly pointed out that there is a gap between preclinical studies aimed at developing approaches to metastatic disease therapy through targeting cell organelles and clinical applications. Further work should be aimed, among other things, at eliminating these limitations. In the conclusion, the authors pointed out the great prospects for using combination therapy for metastatic disease: targeting organelles in combination with photodynamic therapy and immunotherapy.
The article is interesting, timely and important for clinical oncology. The text of the article is written clearly. The manuscript did not raise any ethical issues. All references to publications in the References section are necessary and correct, written in the correct style. Of the 171 References, 60 References are from the last 5 years.
The figures in the article are necessary for the perception of the text of the article. The quality of the figures and legends to them is excellent. A legend must be provided for the table.
I have no concerns about the similarity of this article with other articles published by the same authors.
The competing interests of the authors do not create bias in the presentation of results and conclusions.
Author Response
Reviewer:
Comments:
The review article "Subcellular Organelle Targeting as a Novel Approach to Combat Tumor Metastasis" is submitted for review. The title of the review article reflects its content. This article addresses the issue of the relationship between the process of tumor metastasis and subcellular organelles. In addition, the authors explain the targeting strategies for each subcellular organelle.
In the Abstract section, the authors pointed out the relevance of metastatic disease therapy. According to the authors, targeting subcellular organelles offers promising opportunities for improving drug delivery and the effectiveness of metastasis therapy. The Abstract section fully corresponds to the content of the article. The "Keywords" presented in the article correspond to the content of the article and are necessary. The introduction briefly presents the mechanisms of metastasis. The authors pointed out the inefficiency of the approaches known in the clinic to treat metastatic disease. The material on the connection between the progression of tumor metastases and the structure and function of each subcellular organelle is very important. The authors formulated the aim of the study clearly. This review delves deeply into tumor metastasis processes and their connection with subcellular organelles. In addition, strategies for targeted drug delivery for each subcellular organelle are covered.
The authors conducted a literature search on the following topics: tumor metastasis and its development mechanisms, modern therapy of metastatic disease, the relationship of the metastatic process with subcellular organelles. The main search results were analyzed and presented in the following sections of the article: "2. Potential connection between subcellular organelles and metastasis", "3. Subcellular drug delivery systems for tumor metastasis therapy", "4. Future prospect and conclusion". All sections are necessary. The authors clearly, logically and consistently present the structure of organelles (nucleus, mitochondria, endoplasmic reticulum, Golgi apparatus) in cells and their role in metastasis. An analysis of possible approaches to organelle regulation for the correction of metastatic disease was carried out. The discussion of all results is correct using a significant number of publications from various independent research groups. Each section presents an intermediate conclusion. All terms are used correctly. The authors rightly pointed out that there is a gap between preclinical studies aimed at developing approaches to metastatic disease therapy through targeting cell organelles and clinical applications. Further work should be aimed, among other things, at eliminating these limitations. In the conclusion, the authors pointed out the great prospects for using combination therapy for metastatic disease: targeting organelles in combination with photodynamic therapy and immunotherapy.
The article is interesting, timely and important for clinical oncology. The text of the article is written clearly. The manuscript did not raise any ethical issues. All references to publications in the References section are necessary and correct, written in the correct style. Of the 171 References, 60 References are from the last 5 years. The figures in the article are necessary for the perception of the text of the article. The quality of the figures and legends to them is excellent. I have no concerns about the similarity of this article with other articles published by the same authors. The competing interests of the authors do not create bias in the presentation of results and conclusions.
R: We sincerely thank the reviewer for recognizing our article's structural coherence, comprehensive organelle-metastasis analysis, and clinically oriented therapeutic strategies. The thoughtful comments help us further improve this review.
Q.1: The authors conducted a literature search on the following topics: tumor metastasis and its development mechanisms, modern therapy of metastatic disease, the relationship of the metastatic process with subcellular organelles. It seems to me that the article would be appropriate to reference the medical research databases in which the authors searched for publications necessary for their review article. Are these PubMed, Scopus and/or Web of Science, and possibly other databases? Were there any limitations when searching for publications?
A.1: Thanks for the comments and we appreciate the thoughtful concern. In this review, there is no limitations when searching for publications. We searched the related literatures in PubMed, Scopus, Web of Science and ScienceDirect. A statement about the involved medical research databases has been added in the introduction part.
“In this review, the related literatures in medical research databases including PubMed, Scopus, Web of Science and ScienceDirect were analyzed.”
Q.2: A legend must be provided for the table.
A.2: Thanks for the comments. A revised legend has been added for the table as follows:
“Table 1: The nuclear, mitochondrial, ER and GA compartment are targeted via different mechanisms. This table summarizes the subcellular organelle targeting delivery systems for metastasis treatments in the recent five years.”

Round 2
Reviewer 2 Report
Comments and Suggestions for Authors
Revision is satisfactory.